# Conditional and unconditional components of aversively motivated freezing, flight and darting in mice

**Jeremy M Trott, Ann N Hoffman, Irina Zhuravka, Michael S Fanselow\***

Staglin Center for Brain and Behavioral Health, Department of Psychology, Department of Psychiatry and Biobehavioral Sciences, University of California, Los Angeles, Los Angeles, United States

**Abstract** Fear conditioning is one of the most frequently used laboratory procedures for modeling learning and memory generally, and anxiety disorders in particular. The conditional response (CR) used in the majority of fear conditioning studies in rodents is freezing. Recently, it has been reported that under certain conditions, running, jumping, or darting replaces freezing as the dominant CR. These findings raise both a critical methodological problem and an important theoretical issue. If only freezing is measured but rodents express their learning with a different response, then significant instances of learning, memory, or fear may be missed. In terms of theory, whatever conditions lead to these different behaviors may be a key to how animals transition between different defensive responses and different emotional states. In mice, we replicated these past results but along with several novel control conditions. Contrary to the prior conclusions, running and darting were primarily a result of nonassociative processes and were actually suppressed by associative learning. Darting and flight were taken to be analogous to nonassociative startle or alpha responses that are potentiated by fear. Additionally, associative processes had some impact on the topography of flight behavior. On the other hand, freezing was the purest reflection of associative learning. We also uncovered a rule that describes when these movements replace freezing: when afraid, freeze until there is a sudden novel change in stimulation, then burst into vigorous flight attempts. This rule may also govern the change from fear to panic.

**\*For correspondence:**
fanselow@psych.ucla.edu

## Editor's evaluation

This paper will be of interest to neuroscientists, learning theorists, and clinicians concerned with factors influencing threat-related response selection relevant to fear vs panic. The manuscript describes a group of well-designed experiments that investigate whether flight-like behaviors reported by other groups require associative learning in order to occur. The authors demonstrate that flight-like behaviors observed in these tasks are largely the result of non-associative processes.

## Introduction

Fear limits the behaviors available to an animal to its species-specific defense reactions (SSDRs), thereby precluding more flexible voluntary behavior (*Bolles, 1970*). This characteristic is one reason that conditions characterized by high fear levels such as anxiety disorders are so maladaptive (*Fanselow, 2018*). It is also one reason that Pavlovian fear conditioning is so easy to measure in the laboratory, one can simply measure innate defensive responses (i.e. SSDRs) to diagnose fear and fear-related memory. This has made fear conditioning one of the major rodent assays of learning, memory, and anxiety disorders. Over the last four decades, fear conditioning studies have extensively used one

of these defensive behaviors, freezing, more than any other response (*Anagnostaras et al., 2010*; *Bouton and Bolles, 1980*; *Do-Monte et al., 2015*; *Fanselow and Bolles, 1979*; *Grewe et al., 2017*; *Kim and Fanselow, 1992*; *Kwon et al., 2015*; *Nader et al., 2000*; *Roy et al., 2017*). Freezing is a common and adaptive defensive behavior as it reduces the likelihood of detection and attack by a predator (*Fanselow and Lester, 1988*).

However, if rodents have multiple defensive responses, an important theoretical question is what are the conditions that select between different SSDRs (*Fanselow, 1997*). An influential model of SSDR selection applied to both humans and rodents is Predatory (or Threat) Imminence Continuum theory, which states that qualitatively distinct defensive behaviors are matched to the psychological distance from physical contact with a life-threatening situation (*Bouton et al., 2001*; *Fanselow and Lester, 1988*; *Mobbs, 2018*; *Mobbs et al., 2007*). Stimuli that model particular points along this continuum elicit behaviors appropriate to that level of predatory imminence. For example, rodents freeze when they detect a predator but show vigorous bursts of activity to contact by the predator (*Fanselow and Lester, 1988*). The former, labeled post-encounter defense, relates to fear-like states. The latter, referred to as circa-strike defense, relates to panic-like states (*Bouton et al., 2001*; *Perusini and Fanselow, 2015*). According to this account, in fear conditioning experiments the shock unconditional stimulus (US) models painful contact with the predator and therefore invariably produces circa-strike activity bursts but not freezing (*Fanselow, 1982*). On the other hand, stimuli associated with shock such as an auditory conditional stimulus (CS), model detection of a predator and therefore invariably produce post-encounter freezing as a conditional response (CR) but not activity bursts (*Fanselow, 1989*).

Recently, there have been reports that challenge this view. *Fadok et al., 2017* used a unique two-component serial CS consisting of a 10 s tone followed immediately by a 10 s white noise ending with a 1 s shock and found that the initial component (tone) produced freezing, while the second component (noise) produced bursts of locomotion and jumping in mice. *Gruene et al., 2015* reported that in rats a tone CS resulted in a similar burst of locomotion, labeled darting, and this was replicated in subsequent studies (*Colom-Lapetina et al., 2019*; *Mitchell et al., 2021*). The results were interpreted as a competition between 'active' and 'passive' defenses. These findings not only challenge the above response selection rule but also call for a 'reinterpretation of rodent fear conditioning studies' because if only one SSDR is measured (e.g. freezing) but the situation is characterized by a different SSDR, fear and fear-related learning may be misdiagnosed (*Gruene et al., 2015*). Also note that contrary to Predatory Imminence Theory, *Gruene et al., 2015* suggested that freezing and darting were competing CRs to the same level of threat (*Fanselow, 1989*).

Both previous reports concluded that these activity bursts were CRs because they increased over trials during acquisition when CS and US were paired and decreased during extinction when the CS was presented alone (*Fadok et al., 2017*; *Gruene et al., 2015*). While these behavioral patterns are certainly properties of a CR, they are not diagnostic of associative learning as these changes could also result from nonassociative processes such as sensitization and habituation (*Rescorla, 1967*). Additionally, no formal assessment was made of what properties of the CS led to the alternate CRs (e.g. its serial nature, the ordering of the two sounds, or stimulus modality). One subsequent study using this serial conditioning procedure in mice has suggested that this white-noise-elicited activity burst is mainly a result of the stimulus salience or intensity of the white noise and does not depend on any particular temporal relation to the US (*Hersman et al., 2020*). Another recent study using this procedure in rats has suggested that this flight behavior only occurs in contexts in which fear has been established and is a result of associative processes (*Totty et al., 2021*). Therefore, to better understand the associative nature of these flight responses, we embarked on a series of experiments to test these theoretical views and assess the validity of these concerns (*Tables 1–4*).

## Results

### Experiment 1

Experiment 1 was conducted as delineated in *Table 1* (see *Figure 1* for a schematic representation of the serial conditional stimulus and the design for training and testing for Experiment 1). The first condition was a nearly exact replication of the conditions used by *Fadok et al., 2017*, using male and female mice (Replication Group). Briefly, animals received 10 pairings of footshock and the

**Table 1.** Design of Experiment 1.

| Group | Training treatment: 10 CS-US pairings (5 per day) | Testing treatment (16 on 1 day) |
| --- | --- | --- |
| (1) Replication | 10 s tone→10 s noise→1 s shock | 10 s tone→10 s noise |
| (2) CS duration | 10 s noise→shock | 10 s noise |
| (3) Stimulus change | 20 s tone→shock | 10 s tone→10 s noise |

two-component stimulus (10 s tone followed by 10 s white noise) over 2 days before being tested on the third day with the two-component stimulus (see *Figure 1—figure supplement 1* to view example velocity traces from day 1 training for one mouse in the Replication Group). We scored bursts of locomotion and jumping with a peak activity ratio (PAR; *Fanselow et al., 2019*) and the number of darts (*Gruene et al., 2015*). PAR reflects the largest amplitude movement made during the period of interest, while darts reflect the frequency of large movements during the same period (see Methods). We included two additional groups in this experiment to test the nature of any observed behaviors. We first asked whether any observed behavior occurred to the noise specifically because it was embedded in a serial compound and/or because of the brevity of the noise (10 s). For this group, we simply conditioned and extinguished a 10 s white noise (CS Duration Group). In a third group of mice, we also asked whether the noise-elicited flight behavior required the noise to be present during training. These mice were trained with a 20 s tone, but tested with the two-component serial compound stimulus (Stimulus Change Group).

In the nearly exact replication of the conditions used by *Fadok et al., 2017*, using male and female mice, we obtained nearly identical results with our Replication Group (*Table 1*, *Figure 2*). For this and all experiments described below, no effects of sex were observed in initial comparisons/ANOVAs (see Discussion). Sex was thus removed as a factor in subsequent statistical analyses. In the Replication Group, freezing to the initial tone progressively increased over the course of conditioning. At the beginning of training, freezing increased to the white noise but plateaued after a few trials. When freezing plateaued the noise elicited activity bursts, and this pattern maintained throughout acquisition and the beginning of extinction testing. As extinction testing continued, freezing was maintained while PAR and darting to the noise decreased.

Then, we directly asked whether the plateau in freezing and increase in activity that occurred to the noise required the noise to be a component of a serial compound stimulus. We simply conditioned and extinguished a 10 s white noise (CS Duration Group) and found that freezing increased linearly during a 10 s pre-noise period reflecting the acquisition of contextual fear conditioning (*Kim and Fanselow, 1992*; *Figure 2—figure supplement 1*). During testing, the reaction to onset of the white noise was almost a duplicate to what we saw when the noise was embedded in the compound. In other words, activity bursts and darting in no way depended on the use of a serial compound.

To further probe the necessity of the compound and the presence of the noise during acquisition/shock pairings, we trained a third group of mice with a 20 s tone instead of the compound but tested them with the serial compound stimulus (Stimulus Change Group). During these shock-free tests the noise evoked a very similar PAR and darting behavior to when training was with the compound (*Figure 2*). What is striking about this finding is that even though the noise was never paired with shock it still evoked an activity burst. While behavior at test generally looks very similar for

**Table 2.** Design of Experiment 2.

| Group | 2-Day training treatment: | Testing treatment |
| --- | --- | --- |
| (1) Pseudoconditioned noise-shock only noise test | 10 Shocks (1-mA, 1 s, 150–210 s intertrial interval) | 5 Noise presentations (10 s) |
| (2) Pseudoconditioned tone-shock only tone test | 10 Shocks (1-mA, 1 s, 150–210 s intertrial interval) | 5 Tone presentations (10 s) |
| (3) No shock control | Context exposure only (17 min and 15 s per day) | 5 Noise presentations (10 s) |
| (4) Noise-shock conditioning | 10 Noise (10 s)→shock pairings | 5 Noise presentations (10 s) |

**Table 3.** Design of Experiment 3—paired vs unpaired noise-shock.

| Group | Training treatment: 10 CS-US pairings (5 per day) | Testing treatment (2 on 1 day) |
|---|---|---|
| (1) Paired noise-shock (conditioning) | 10 s noise→1 s shock | 10 s noise |
| (2) Unpaired noise-shock | 10 s noise and 1 s shock – unpaired | 10 s noise |
| (3) Noise - CS only | 10 s noise | 10 s noise |
| (4) Shock only (pseudo conditioning) | 1 s shock | 10 s noise |

the Stimulus Change and Replication groups, direct statistical comparisons reveal some minor differences. For PAR, a repeated measures ANOVA with Trial, Group, and CS type revealed a main effect of CS type ($F_{[1, 24]}$=53.121, p<0.001) as well as a Trial X Group interaction ($F_{[15, 360]}$=1.970, p=0.017) such that Stimulus Change animals showed greater PAR on trials 2 and 5 (p's=0.003, 0.05). Further, for only the Stimulus Change Group, PAR decreased over the session (p=0.034). For darting, a repeated measures ANOVA with Trial, Group, and CS type revealed a main effect of group ($F_{[1, 24]}$=4.321, p=0.048) such that the Stimulus Change Group darted more than the Replication Group, with no interaction with CS type. For both PAR and darting, despite unreliable interactions with CS type in the ANOVAs above, additional ANOVAs were run on responding for each CS type individually to assess potential impacts of group on tone vs noise responding specifically. In each of these ANOVAs, there were no reliable effects of group (all p's>0.05). Thus, if there are any differences between Replication and Stimulus Change groups, it is that the Stimulus Change Group (no noise-shock pairing) generally shows more flight than the Replication Group (noise-shock pairing). These findings strongly implicate nonassociative processes in the activity burst rather than conditioning.

Overall in Experiment 1, we replicated findings that different defensive behaviors develop to separate components of a serial CS (Replication Group). This pattern of behavior holds true if the noise is presented by itself during training (CS Duration Group), and this pattern of behavior at testing does not require the noise to be present during training (Stimulus Change Group). Despite differences in behavioral procedures used across acquisition and extinction, we sought to examine any differences in reactivity to the noise during extinction testing between these three groups. We directly analyzed velocity data across the three groups (*Figure 3*). We focused on the first four trials of extinction testing as this is when the majority of the darting behavior occurred, and we further narrowed our analyses to the 10 s noise period as all groups received at least the 10 s noise at test.

A mixed model ANOVA revealed a significant effect of time ($F_{[19, 361]}$=8.203, p<0.001) as well as a group X time interaction ($F_{[38, 361]}$=1.497, p=0.034). Generally, velocity peaked during the first bins of the noise period and then quickly decreased to more stable levels. Post hoc analyses revealed that the Stimulus Change Group trended to have the elevated velocity during the first bin of the noise period with trends for higher velocity than the CS Duration Group (p=0.09) and did have significantly higher velocity than the CS Duration Group during the fifth bin (~2.5 s into the noise; p=0.04).

**Table 4.** Design of Experiment 4—tested the effect of habituation to the white noise.

| Group | Habituation treatment: 10 CS exposures (5 per day) | Training treatment: 10 CS-US pairings (5 per day) | Testing treatment (3 on 1 day) |
|---|---|---|---|
| (1) Habituation/shock only (H-shock) | 10 s noise | 1 s shock | 10 s noise |
| (2) Habituation/paired noise-shock (H-paired) | 10 s noise | 10 s noise→1 s shock | 10 s noise |
| (3) Context exposure/shock only (C-shock) | Context exposure | 1 s shock | 10 s noise |
| (4) Context exposure/paired noise-shock (C-paired) | Context exposure | 10 s noise→1 s shock | 10 s noise |

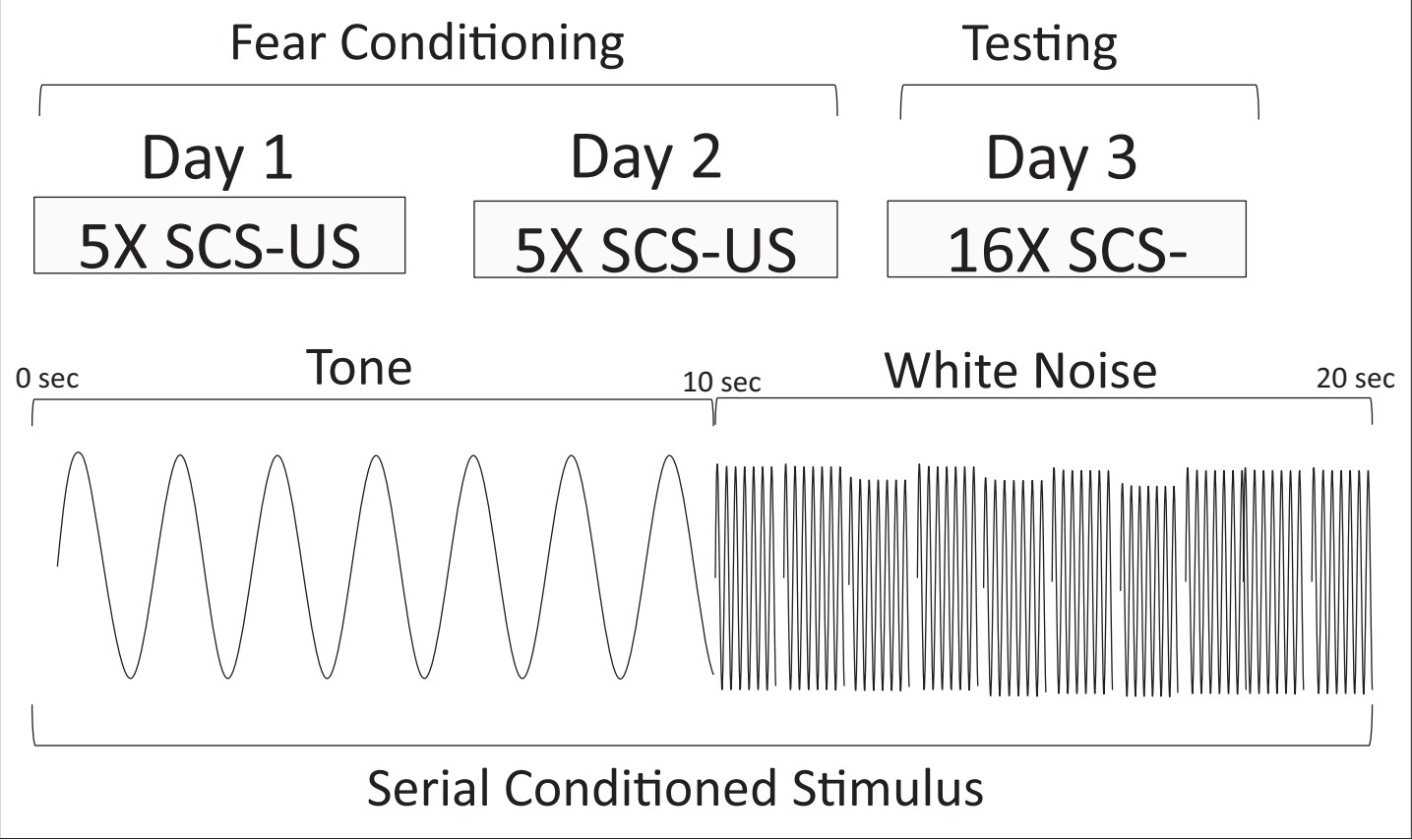

**Figure 1.** Behavioral design and schematic representation of the serial compound conditional stimulus (SCS) used for the Replication Group in Experiment 1. During training, animals were given 2 days each of five SCS-US pairings. The SCS consisted of a 10 s pure tone (7.5 kHz) followed by a 10 s white noise (75 dB). Immediately upon termination of the white noise-SCS, a footshock US (1 s, 0.9 mA) was delivered. On day 3, the animals were tested with 16 presentations of the SCS without delivering any shocks.

The online version of this article includes the following source data and figure supplement(s) for figure 1:

**Figure supplement 1.** Example traces of velocity (cm/s) measurements obtained via EthoVision across five trials on the first day of training for a mouse in the Replication Group of Experiment 1.

**Figure supplement 1—source data 1.** Source files for velocity used to create representative traces.

While the noise did not need to be within a serial compound stimulus or even need to be presented during training in order to elicit flight, it is worth noting that the strongest noise-elicited flight occurred for the group that received the serial compound stimulus only at test and for which the noise was novel at test.

## Experiment 2

The mice that received the 20 s tone during training but the compound during testing showed darting to the noise embedded in the compound (*Figures 2 and 3*). Since the noise was not paired with the shock, this suggests that the response to the noise was nonassociative. However, it is possible that during the initial test trials the response to the noise occurred via second-order conditioning as the noise was paired with the previously reinforced tone. This seems unlikely because most darts were seen at the beginning of testing and decreased over the session. A second-order conditioning interpretation suggests the opposite pattern. Nonetheless, in a second experiment, we included classic controls to directly test for the phenomenon of pseudoconditioning (*Table 2*). Pseudoconditioning is a form of sensitization whereby mere exposure to the US changes behavior to the stimulus used as a CS (*Underwood, 1966*), and this appears to be what was observed in Experiment 1 (Stimulus Change Group; *Figure 2*). Two pseudoconditioned groups of mice simply received the same shock schedule used in the prior study without any auditory stimuli (no CS). A third was merely exposed to

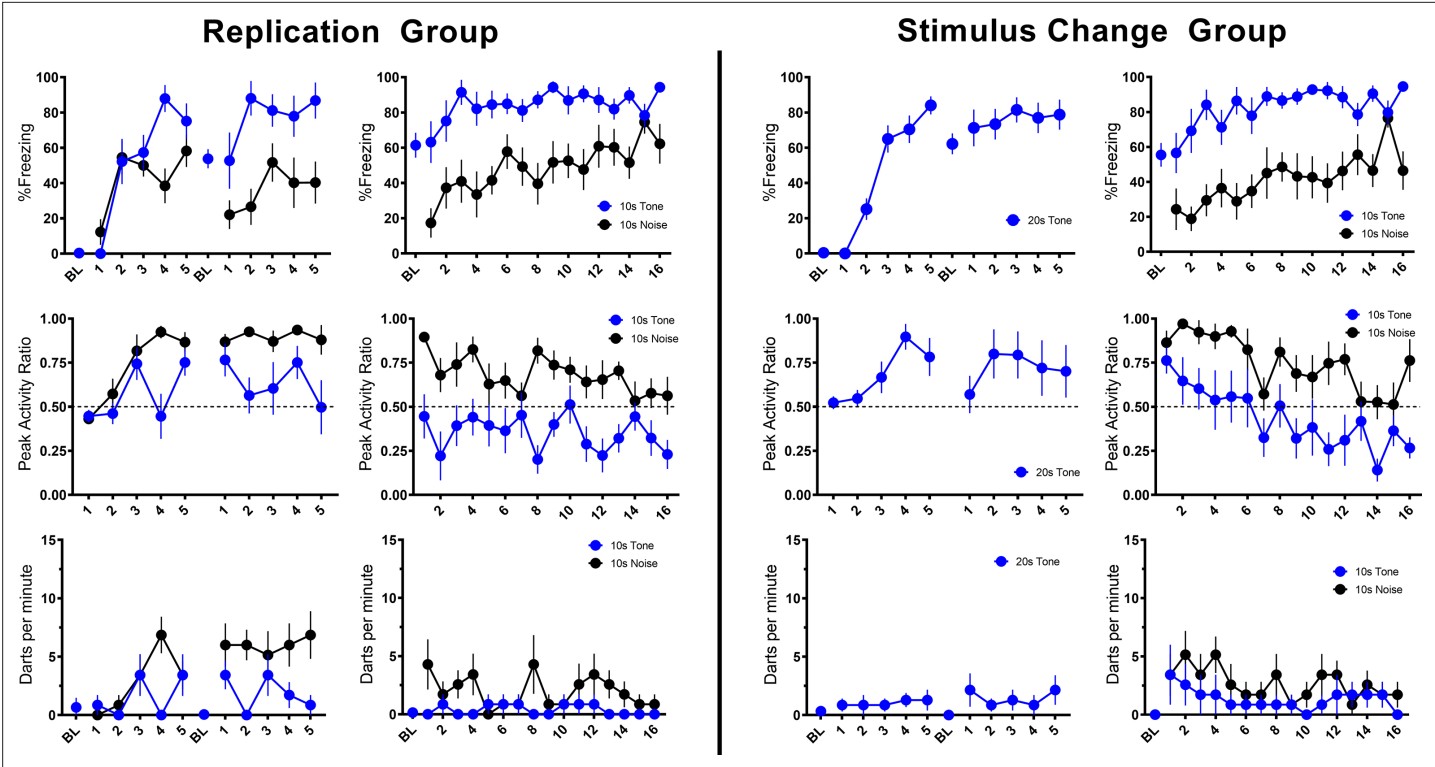

**Figure 2.** Trial-by-trial mean (±SEM) percent freezing, peak activity ratio, and darts per minute throughout all stimulus presentations during training (left panels) and testing (right panels) for the Replication Group (n=7; left half of figure) and the Stimulus Change Group (n=7; right half of figure) in Experiment 1. See *Figure 2—source data 1* & *Figure 2—source data 2*.

The online version of this article includes the following source data and figure supplement(s) for figure 2:

**Source data 1.** Source files for freezing, PAR, and darting for Experiment 1-Replication and Stimulus Change groups.

**Source data 2.** Source files for velocity for all of Experiment 1.

**Figure supplement 1.** Mean (±SEM) percent freezing, peak activity ratio, and darts per minute throughout training (left panels) and testing (right panels) for the CS Duration Group (n=8) of Experiment 1.

**Figure supplement 1—source data 1.** Source files for freezing, PAR, and darting for Experiment 1-CS Duration group.

the chamber. The final group was a conditioning group that received noise-shock pairings. All groups received tests with the 10 s noise, except for one of the pseudoconditioning groups that was tested with the tone.

*Figures 4 and 5* summarize the test results from Experiment 2 (see *Figure 4—figure supplement 1* for trial-by-trial data). As would be expected for a CR, freezing to the noise was greatest in the mice that received noise-shock pairings (F[3, 28]=11.76, p<0.001). Significant associative learning was indicated by more noise-elicited freezing in the paired group than the shock-only trained group tested with the noise. Interestingly, the no shock group that was tested with the noise gradually increased freezing over the course of noise testing (*Figure 4—figure supplement 1*) suggesting that the 75 dB noise itself was aversive to the mice and could support some conditioning of freezing (i.e. it was a weak US).

The test session data were very different for activity bursts (*Figures 4 and 5*). The greatest PAR occurred in the pseudoconditioned control (shock only during training) that was tested with the novel noise (F[3, 28]=20.085, p<0.001). The pseudoconditioned control tested with the novel noise also showed the most darting behavior. Furthermore, these results are supported by a direct analysis of velocity data during the 10 s CS period at test (*Figure 5*).

A mixed model ANOVA on the averaged velocity measures during the CS period for the first four trials of the test session revealed significant effects of group (F[3, 28]=5.796, p=0.003) and time (F[4.06, 113.69]=6.038, p<0.001) as well as a group X time interaction (F[12.18, 113.69]=2.695, p=0.003). Generally, velocity again peaked during the first bins of the noise period and then quickly

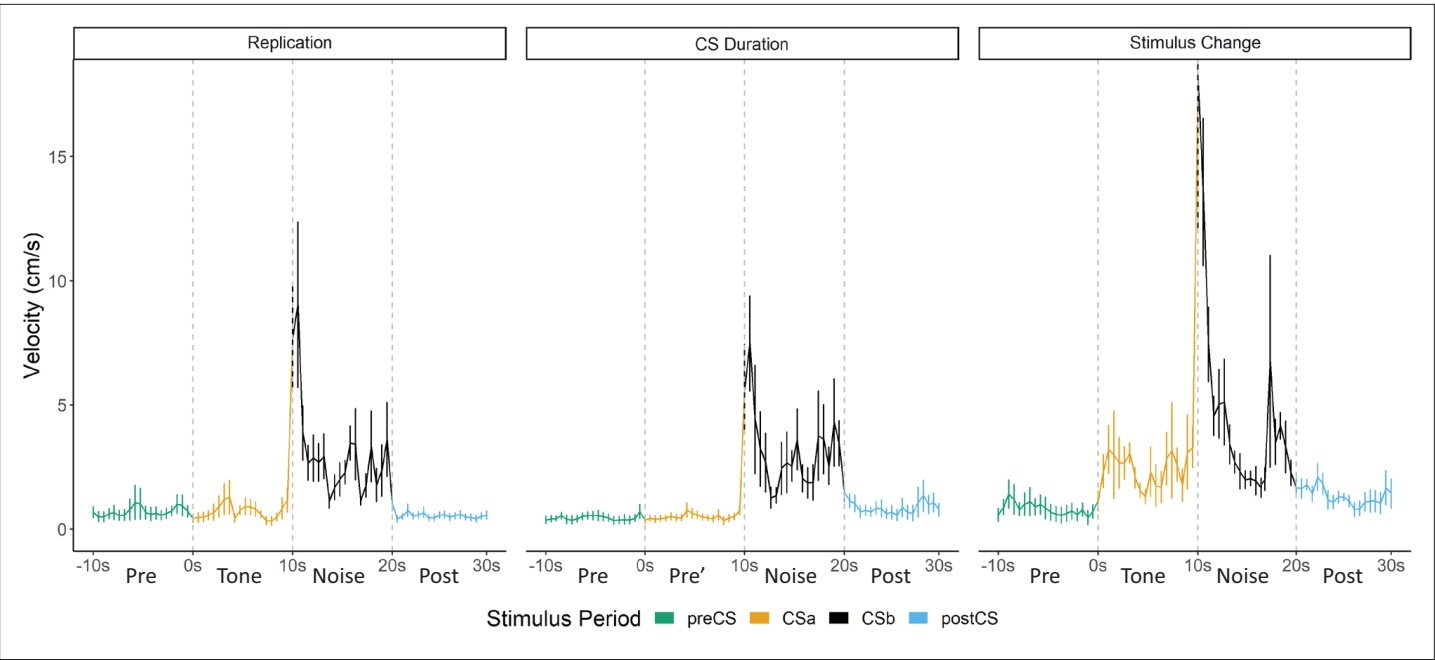

**Figure 3.** Averaged traces of velocity (cm/s) across the first four trials of extinction during testing for Experiment 1. Data were averaged across all animals per group and binned into ~0.5 s bins (0.533 s) and presented as mean ± SE. These within-subject error bars are corrected for between-subject variability using methods as described in *Morey, 2008*. During this test, the Replication Group (n=7) and the Stimulus Change Group (n=7) received the serial conditional stimulus in which a 10 s tone was followed by a 10 s noise. The CS Duration Group (n=8) was only tested with a 10 s noise. See *Figure 3—source data 1*.

The online version of this article includes the following source data for figure 3:

**Source data 1.** Source files for velocity for test day of Experiment 1.

decreased to more stable levels. Post hoc analyses revealed that the shock only-noise test group had the highest velocity during the second bin of the noise period (the first second of the CS) with significantly higher velocity than the no shock-noise test (p=0.03), shock only-tone test (p=0.004) and, importantly, the noise shock-noise test groups (p=0.007).

Pseudoconditioning is indicated by more activity during the noise test in the previously shocked mice than the no-shock controls tested with the same noise. Note that for both of these groups the noise was novel during testing so it had no association with shock. It is worth noting that we also see indirect evidence of pseudoconditioning to the tone, such that the shock only tone test group does show an elevated PAR with respect to the nonshocked controls tested with the noise. While darting was very low in this group, it was not zero, suggesting that a novel pure tone stimulus may also support cue-elicited flight behavior in frightened animals, although to a lesser extent than a white noise stimulus, which may have inherently aversive properties (continued in Discussion). Another striking finding is that while the group that received noise-shock training showed an elevated PAR, the level was significantly less than the pseudoconditioning control (p<0.001). Not only are activity bursts not conditioned, these data suggest conditioning may actually suppress such activity bursts. In other words, flight and darting are primarily a result of nonassociative processes and are likely not CRs.

## Experiment 3

In a third experiment, we included a control group in which the shock and noise were explicitly unpaired to again test for the phenomenon of pseudoconditioning but in a situation where exposure to the CS is equated during training (*Table 3*). One group was again a conditioning group that received noise-shock pairings, and one group was again a pseudoconditioned group that only received shocks without any CS. A third group received equal numbers of noise and shock presentations but in an explicitly unpaired manner. An additional control group received presentations of only the white noise CS to examine whether or not the CS alone was able to support conditioning and/or activity bursts.

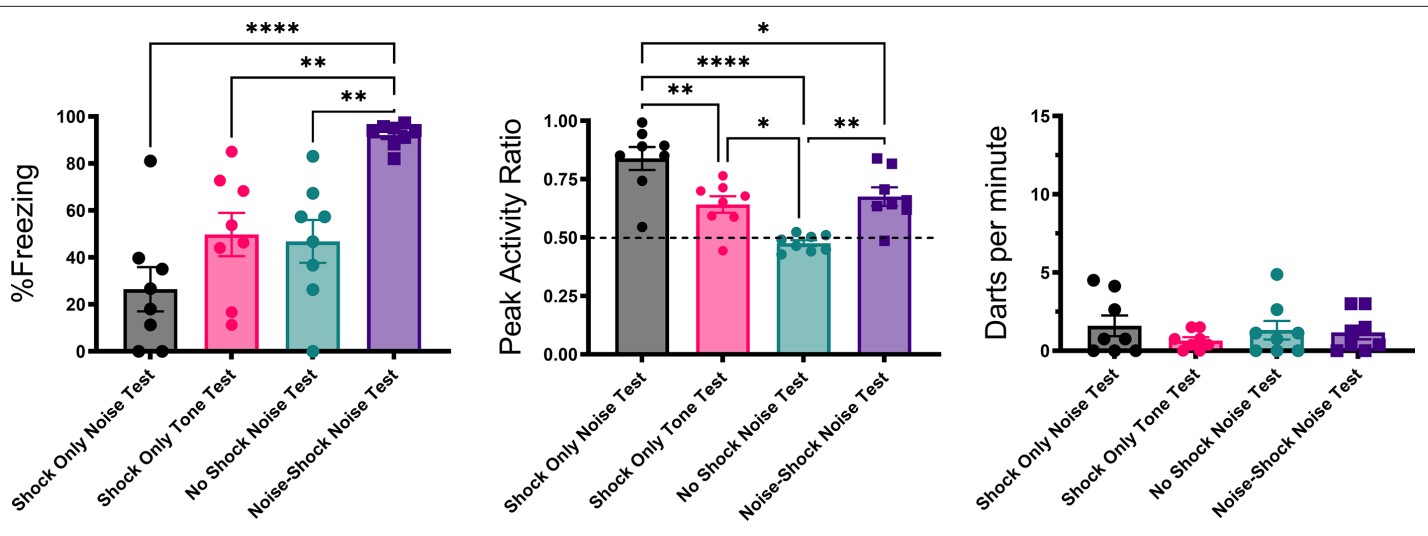

**Figure 4.** Mean (± SEM) percent freezing, peak activity ratio, and darting for the test session for Experiment 2 (n=8 per group). Values are averaged across the 16 trials of extinction during test. p-values and significance were determined through one-way ANOVA. *p<0.05, **p<0.01, ****p<0.0001. See *Figure 4—source data 1* & *Figure 4—source data 2*.

The online version of this article includes the following source data and figure supplement(s) for figure 4:

**Source data 1.** Source files for freezing, PAR, and darting for test day for Experiment 2.

**Source data 2.** Source files for velocity for Experiment 2.

**Figure supplement 1.** Trial-by-trial mean (± SEM) percent freezing, peak activity ratio, and darts per minute throughout 16 trials of testing for Experiment 2 (n=8 per group).

**Figure supplement 1—source data 1.** Source files for freezing, PAR, and darting for Experiment 2.

Acquisition and test results are summarized in *Figures 6 and 7*. As seen in the prior experiments, across training freezing to the white noise rose, and then plateaued in the paired and unpaired groups, at which point the noise began to elicit activity bursts. In the CS only group white noise alone supported low, but consistent levels of freezing but in the shocked groups the noise disrupted freezing to the context. During training, the paired and unpaired groups showed elevated PAR to the noise (F[3, 28]=29.94, p<0.001 for day 1; F[3, 28]=75.18, p<0.001 for day 2), and increased darting to the noise (F[3, 28]=9.392, p<0.001 for day 1; F[3, 28]=29.746, p<0.001 for day 2). Interestingly, for darting, the paired group showed elevated responding on both day 1 (p=0.017) and on day 2 (p=0.004) compared to the unpaired group. During testing, activity bursts (measured as both PAR and darting) to the noise were elevated in all groups that received shock (F[3, 28]=13.35, p<0.001 for PAR; F[3, 28] = 8.160, p<0.001 for darting). Again, similar to training, darting appeared to be the most elevated in the paired group on trial 1 of testing.

While overall darting was elevated in the paired group (during acquisition and on the first trial of testing), the velocity traces during testing (*Figure 7*) reveal that the magnitude/frequency of the initial activity burst to the noise appears to be reduced in the paired group, and that increased levels of activity bursts during the latter portion of the CS account for any differences in overall numbers of darts. Indeed, a direct analysis of the velocity data during the 10 s noise CS period at test revealed significant effects of group (F[3, 28]=9.733, p<0.001), time (F[5.15, 144.22]=9.614, p<0.001), as well as a group X time interaction (F[15.45, 144.22]=2.045, p=0.02). Generally, as seen in prior experiments, velocity again peaked during the first bins of the noise period and then quickly decreased to more stable levels. In the paired group specifically, there is an additional peak of activity in the latter half of the stimulus period. Post hoc analyses revealed that the unpaired group had the highest velocity during the first bin of the noise period (the first second) with significantly higher velocity than the CS only group (p=0.007). Additionally, in the 16th and 17th bins toward the end of the CS period, the paired group showed the most activity with significantly higher velocity than the CS only group (p=0.002 and p=0.001), the shock only group (p=0.001, p=0.02), and the unpaired group (p<0.001, p=0.003).

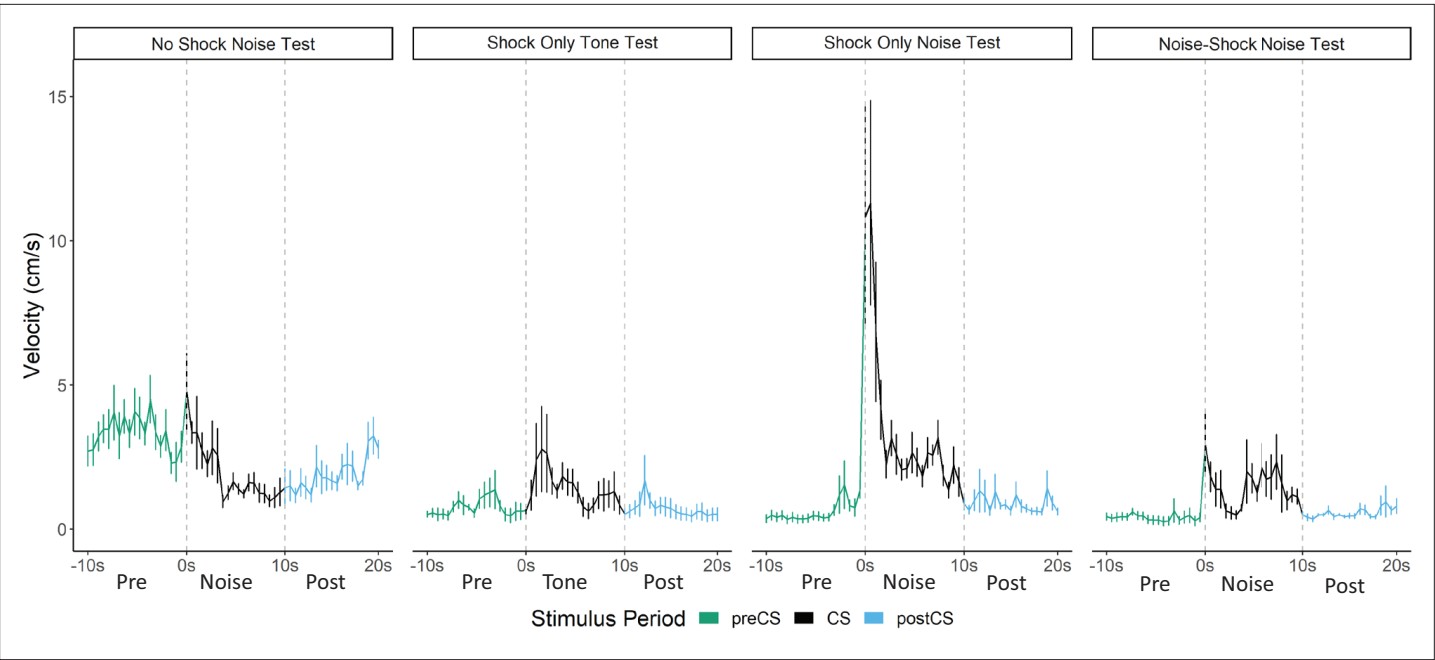

**Figure 5.** Averaged traces of velocity (cm/s) across the first four trials of extinction during testing for Experiment 2 (n=8 per group). Data were averaged across all animals per group and binned into ~0.5 s bins (0.533 s) and presented as mean ± SE. These within-subject error bars are corrected for between-subject variability using methods as described in *Morey, 2008*. During this test, the no shock-noise test, shock only-noise test, and noise-shock noise test groups were tested with a 10 s noise. The shock only-tone test group was tested with a 10 s tone. See *Figure 5—source data 1*.

The online version of this article includes the following source data for figure 5:

**Source data 1.** Source files for velocity for test day of Experiment 2.

That pairing noise and shock altered the timing of the activity bursts is an interesting fact worth considering and suggests that pairing noise and shock may have primarily resulted in a conditional freezing response that in fact competes with/reduces any initial nonassociative activity/bursting to the white noise. Taken together, this and the prior experiment using control groups to assess pseudoconditioning reveal that a large portion of the noise-elicited activity bursts observed is due to nonassociative processes that result in an increase in darting behavior to the noise following shock exposure, regardless of any direct training history of the noise with shock. There does appear to be evidence that pairing noise with shock may further increase or alter the timing of this behavior, but by no means is pairing noise with shock necessary to produce these activity bursts.

## Experiment 4

The experiments thus far have suggested that much of the white-noise-elicited activity bursting is a nonassociative process. We have also shown that novelty of the CS at test may increase this noise-elicited activity (*Figures 3 and 4*). In a final, fourth experiment, we explicitly tested whether habituation to the white noise stimulus prior to noise-shock training would be able to reduce noise-elicited activity bursts. If increased levels of novelty of the CS are driving noise-elicited activity bursts, then prior habituation should reduce the levels of darting to the noise CS. In this experiment, we had four groups that differed in whether they received an additional 2 days of habituation to the white noise stimulus (five noise presentations each day) and whether they received noise-shock pairings during training or just shock only (*Table 4*). One comparison of particular interest was between the habituated and nonhabituated shock only groups as these groups would directly compare whether prior experience with the CS would decrease darting at test compared to a group for which the CS was completely novel.

*Figure 8* shows the results of Experiment 4 during testing (see *Figure 8—figure supplement 1* for trial-by-trial results for freezing, PAR, and darting across habituation, training, and testing). During the 2 days of habituation, interestingly, we found that within groups that received habituation, a low level of darting to the white noise alone without any shock decreased across day 1 ($F_{[4, 48]}=2.887$, p=0.026)

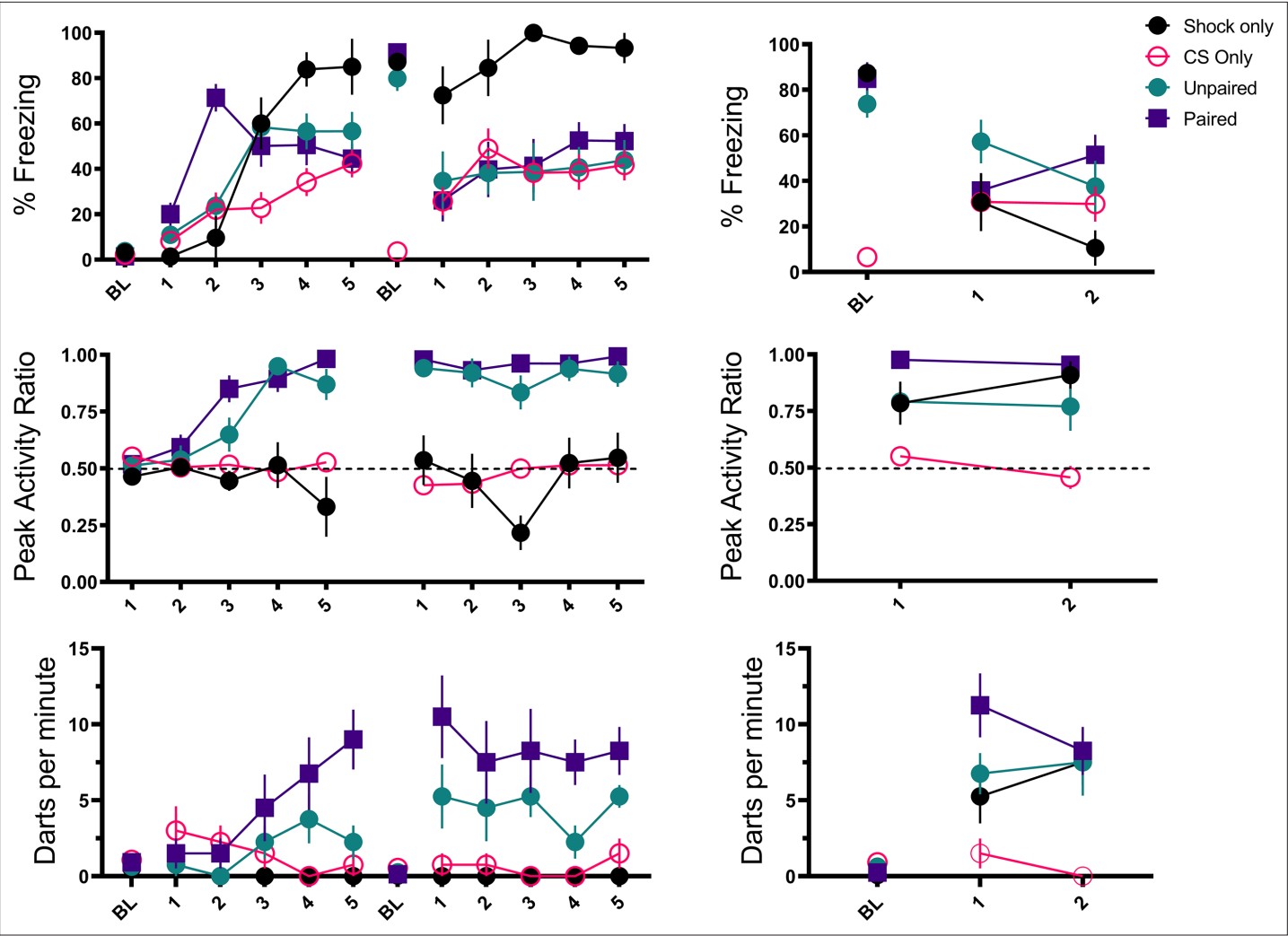

**Figure 6.** Trial-by-trial mean (± SEM) percent freezing, peak activity ratio, and darting per minute throughout all stimulus presentations during training (left panels) and testing (right panels) for Experiment 3 (n=8 per group). See (*Figure 6—source data 1* & *Figure 6—source data 2*).

The online version of this article includes the following source data for figure 6:

**Source data 1.** Source file for freezing, PAR, and darting for Experiment 3.

**Source data 2.** Source files for velocity for Experiment 3.

and increased by the end of the second day of habituation (F[4, 48]=2.793, p=0.36; *Figure 8—figure supplement 1*). Concurrently, freezing to the white noise increased over habituation trials, again showing that this white noise stimulus alone can act as a US. It is interesting that darting occurred to the white noise at the start of habituation when the CS was very novel, and at the end of habituation once the white noise alone was able to support some level of fear.

Comparing the two shock only groups during test, the noise disrupted freezing more than tone. In this regard, noise seems to act like a weak shock US (*Fanselow, 1982*). Like shock, it disrupts freezing (*Figure 4—figure supplement 1*) and like shock, it supports conditioning of freezing (*Figure 6*).

Within paired groups (H-paired and C-paired), we found that throughout acquisition and particularly on the second day of training (*Figure 8—figure supplement 1*), prior habituation to the white noise increased freezing (F[1, 24]=5.701, p=0.025) and decreased noise-elicited darting (F[1, 24]=5.130, p=0.033), as predicted if prior exposure to the CS functions to reduce any partially novelty-driven darting. We again saw that freezing to the white noise initially increased during acquisition, but as the darting response begins to become more apparent, freezing decreases to medium levels. At test (*Figure 8*, *Figure 8—figure supplement 1*), for freezing, we found a main effect of pairing (F[1,

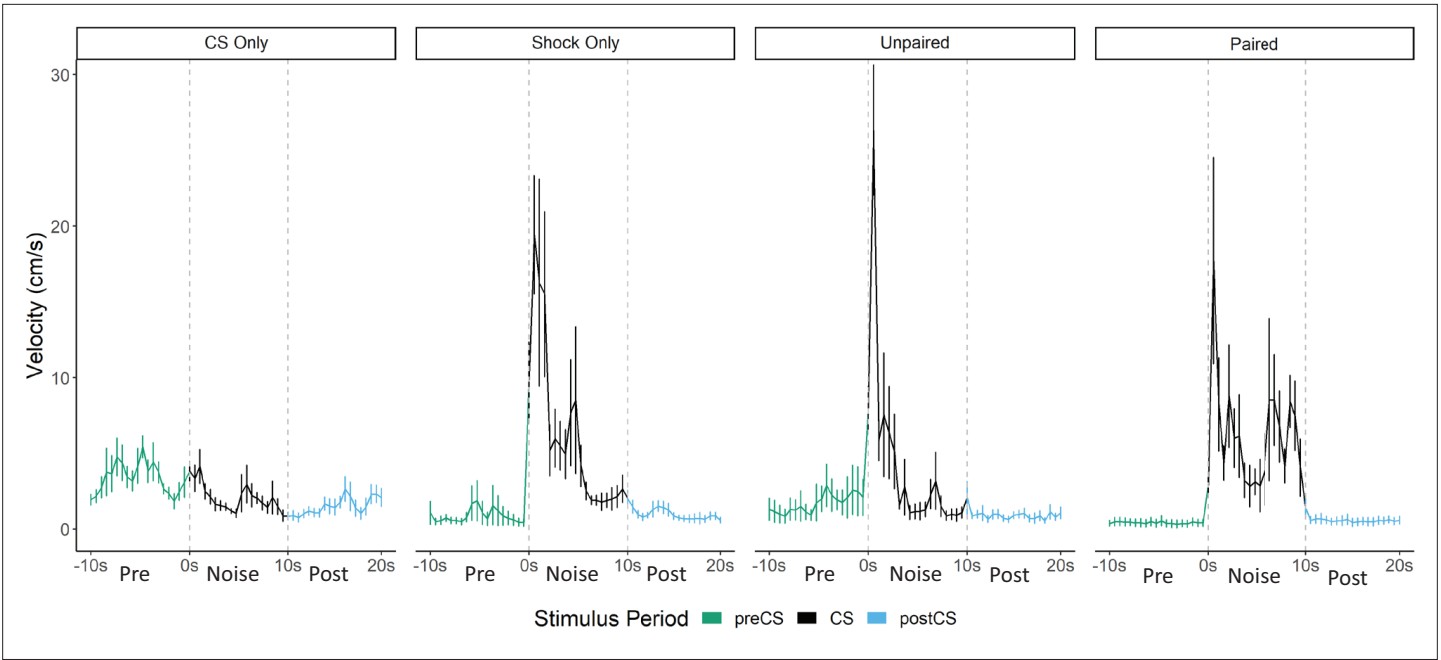

**Figure 7.** Averaged traces of velocity (cm/s) across two trials of extinction during testing for Experiment 3 (n=8 per group). Datawere averaged across all animals per group and binned into ~0.5 s bins (0.533 s) and presented as Mean ± SE. These within-subject error bars are corrected for between-subject variability using methods as described in *Morey, 2008*. During this test all groups were tested with a 10 s noise CS. See *Figure 7—source data 1*.

The online version of this article includes the following source data for figure 7:

**Source data 1.** Source files for velocity for test day of Experiment 3.

24]=11.306, p=0.003), such that animals who received white noise paired with shock froze more than animals who only received shock during acquisition, again indicative that noise-elicited freezing is a conditional behavior that results from associative learning. For darting behavior, we found a habituation X pairing interaction (F[1, 28]=4.939, p=0.035) such that pairing white noise with shock increased darting within habituated animals (p=0.033), and that habituation reduced darting within animals who only received shock during training (p=0.045). These results reveal multiple points of interest. First, and as shown in prior experiments, the white noise acts as a US on its own and needs not be paired with shock to produce darting at test. Merely experiencing the shock is enough to produce darting to the white noise at test (pseudoconditioning due to sensitization). Furthermore, prior experience with the white noise, through habituation, actually reduced this darting at test. Additionally, in this experiment, we do again show evidence that pairing white noise with shock can further increase darting behavior at test, at least within animals who have already experienced the noise during habituation. Again, as with Experiment 3 (*Figure 7*), the timing of the darting response in paired groups is fundamentally altered compared to shock only groups (*Figure 8*). The magnitude/frequency of the initial activity burst to the noise appears to be reduced in the paired groups, and increased levels of activity bursts during the latter portion of the CS account for any differences/increases in overall numbers of darts.

Indeed, a mixed model ANOVA with pairing, habituation, and time as factors on the averaged velocity traces for each trial revealed significant effects of time (F[56, 1568]=17.420, p<0.001), a habituation X pairing interaction (F[1, 28]=4.696, p=0.04), and a pairing X time interaction (F[56, 1568]=3.036, p=0.01). Generally, once again, velocity peaked during the first bins of the noise period and then quickly decreased to more stable levels. As seen in the experiments above, again, this initial peak in velocity was most apparent in the shock only groups, with the paired groups showing an initially smaller peak in velocity. Post hoc analyses revealed that the shock only groups had significantly higher velocity during the first three bins of the noise than the paired groups (p's=0.02, 0.03, 0.005, respectively). Post hoc analysis on the pairing X habituation interaction reveals that within the nonhabituated groups, pairing noise and shock significantly reduced the velocity throughout test trials (p<0.001). Additionally, within shock only groups, habituation reduced the velocity throughout test

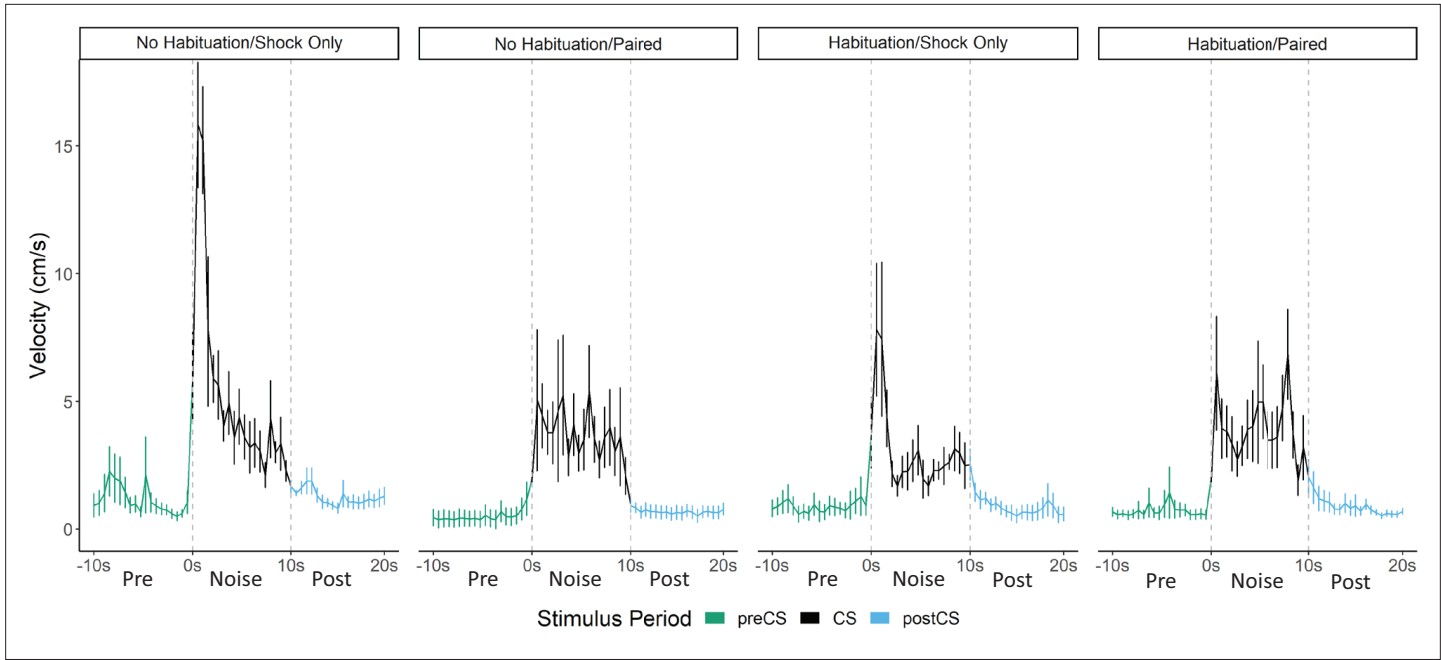

**Figure 8.** Averaged traces of velocity (cm/s) across three trials of extinction during testing for Experiment 4 (n=8 per group). Datawere averaged across all animals per group and binned into ~0.5 s bins (0.533 s) and presented as mean ± SE. These within-subject error bars are corrected for between-subject variability using methods as described in *Morey, 2008*. During this test, all groups were tested with a 10 s noise CS. See *Figure 8—source data 1*.

The online version of this article includes the following source data and figure supplement(s) for figure 8:

**Source data 1.** Source files for velocity on test day of Experiment 4.

**Figure supplement 1.** Trial-by-trial mean (± SEM) percent freezing, peak activity ratio, and darting per minute throughout all stimulus presentations during habituation (left panels), training (middle panels), and testing (right panels) for Experiment 4 (n=8 per group).

**Figure supplement 1—source data 1.** Source files for freezing, PAR, and darting for all of Experiment 4.

**Figure supplement 1—source data 2.** Source files for velocity for all of Experiment 4.

trials (p<.001). These results are exactly what would be predicted if exposure to the noise CS (through pre-exposure and/or through pairing CS and US) in fact reduces noise-elicited activity bursts and flight/darting behavior, that is, darting is enhanced by novelty.

## All experiments analysis of dart timing and topography

While the majority of the data presented here suggest that cue-elicited flight or darting is due primarily to nonassociative influences, we do show evidence that associative processes/pairing noise and shock alter the timing/topography of such flight behavior. Thus, we set out to further analyze these differences in dart timing, and in particular, we were interested in whether initial darts at CS onset may be functionally distinct from darts that occur later on in the CS period.

*Figure 9* shows a detailed analysis of darting magnitude and timing collapsed across all experiments for all animals that received shock during training. First, *Figure 9A and B* represents the magnitude of darts to the tone and noise stimuli during testing, as well as the reaction to the first shock on day 1 of training. Generally, there was an effect of stimulus on response magnitude (W[2, 123.2]=105.3, p<0.0001). The magnitude of response to shock was greater than to tone (p<0.0001) and to noise (p<0.0001), and darts to the noise tended to be stronger than darts to the tone (p=0.043). *Figure 9C* shows the average magnitude of darting responses when an individual animal performs two darts within one stimulus (total n=65 'multi-darts'). On average, within a given single CS presentation, the magnitude of the response for the first dart was greater than for the second dart (t=2.641, df = 64, p=0.01). The magnitude of darts that occurred during the initial 3 s of the 10 s CS period and those that occurred during the final 7 s of the 10 s CS period are shown in *Figure 9D* (by group and stimulus) and *Figure 9E* (collapsed across groups and stimulus). An omnibus ANOVA with group and stimulus

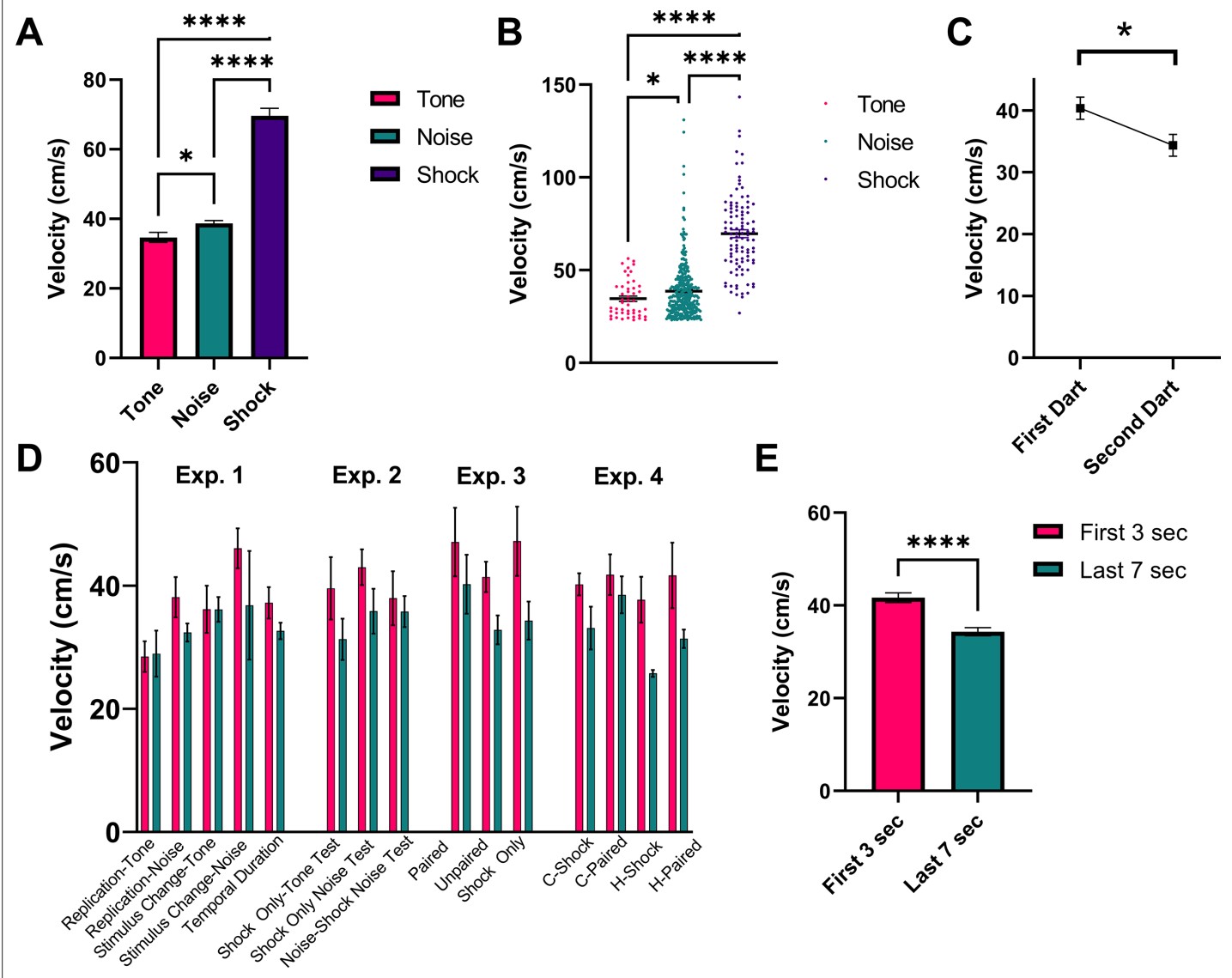

**Figure 9.** Analysis of dart timing and magnitude. (**A**) and (**B**) represent the magnitude of darts to the tone (n=48 darts) and noise (n=360 darts) stimuli during testing, as well as the reaction to the first shock (n=102 shocks) on day 1 of training. Data are presented as mean ± SE and come from all groups (total n=102 animals) that received shock during training, collapsed across all experiments. p-values and significance were determined through Welch's ANOVA. *p<0.05, ****p<0.0001. (**C**) represents the magnitude (mean ± SE) of the first and second dart within a single CS presentation for all animals across all experiments that performed two darts within a single 10 s CS period (n=65 "multi-darts"). p-values and significance were determined through a paired sample t-test. *p<0.05. (**D**) and (**E**) represent the magnitude of darts that occurred during the initial 3 s of the 10 s CS period (n=230 darts) and those that occurred during the final 7 s of the 10 s CS period (n=178 darts). Data are presented as mean ± SE and come from all groups that received shock during training (n=102 animals), displayed by group and stimulus type in (**D**) and collapsed across all experimental groups in (**E**). p-values and significance were determined through Welch's ANOVA. ****p<0.0001. See *Figure 9—source data 1*.

The online version of this article includes the following source data for figure 9:

**Source data 1.** Source files for dart magnitudes and shock reactivity across all experiments.

period (early vs late) revealed a significant effect of stimulus period (F[1, 331]=16.23, p<0.0001) with no effects of/interaction with group. Darts that occurred early during the initial CS onset were larger in magnitude than those that occurred later in the session, suggesting that these two responses may in fact be distinct types of flight behavior (see Discussion).

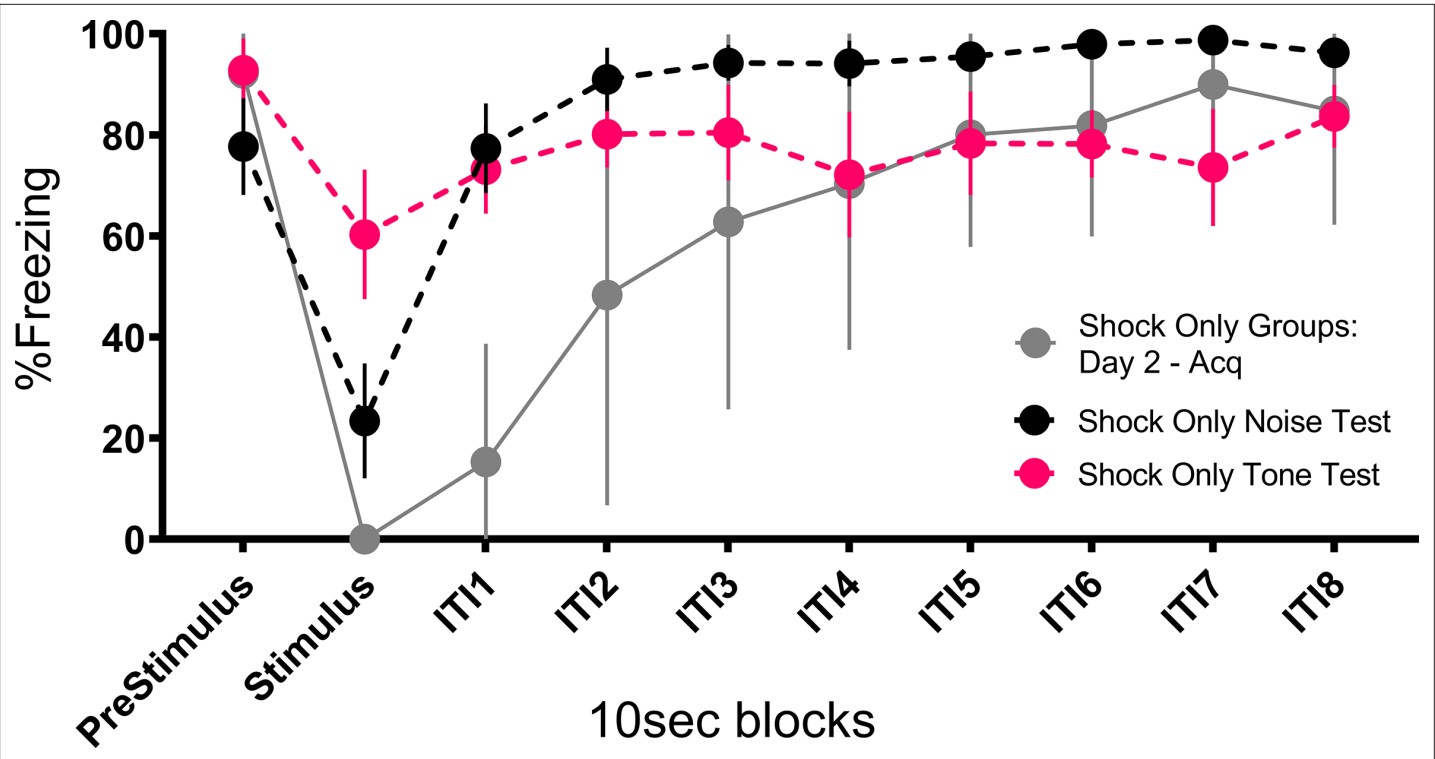

**Figure 10.** Mean (± SEM) percent freezing during extinction/testing for Experiment 2 shows that the occurrence of the stimuli at test disrupt freezing to the context and that the noise disrupts freezing to a greater extent than the tone (n=8 per group). Also plotted is a similar curve showing freezing and the impact that shock presentation during training has on freezing. These data are averaged across both shock only groups (total n=16) on day 2 trials after fear to the context had been established, showing that shock disrupts freezing to an even greater extent than the noise. See *Figure 10—source data 1*.

The online version of this article includes the following source data for figure 10:

**Source data 1.** Source data for the ability of cues and shock to disrupt freezing.

## Discussion

Prior work reported that contact/pain-related stimuli (e.g. shock) disrupt post-encounter freezing and provoke panic-like circa-strike defensive behaviors (*Fanselow, 1982*). The current results suggest a modification of the rules governing a transition between these behavioral states. The rule is that when you are in the post-encounter mode (fear) a sudden change in stimulation, particularly the onset of an intense novel stimulus, can cause an immediate transition to the circa-strike mode (panic). Indeed, the vast majority of the activity bursts/darting behavior occurred at the onset of the stimulus (*Figures 3, 5, 7 and 8*). The effectiveness of this transition depends on the qualities of the stimulus. Stronger shocks cause a greater disruption of freezing and a longer activity burst, yet the same stronger shocks simultaneously condition more freezing to the prevailing cues (*Fanselow, 1982*). The current data call for an expansion of this rule to non-nociceptive stimuli. Like shock, both tone and noise disrupted ongoing freezing, the noise did so for longer than the tone (*Figure 10*), and noise on its own was able to support a minimal level of fear conditioning (*Figure 6*, *Figure 4—figure supplement 1*, *Figure 8—figure supplement 1*). The rule is: when in a state of fear (post-encounter defense) sudden stimulus change provokes panic-like circa-strike defenses proportional to stimulus intensity and novelty.

As the majority of the experiments presented here and in most prior studies conduct both training and testing in the same context (*Fadok et al., 2017*; *Gruene et al., 2015*; *Hersman et al., 2020*), these animals would already be in a high state of fear or post-encounter defense (from any learned contextual fear during training), thus endowing the presentation of the white noise to be a particularly startling stimulus change that can provoke these panic-like flight responses. Novelty of the stimuli is an important factor and familiarity with the CS during conditioning and/or habituation reduced CS novelty for the test. In the experiments presented here, the mice that received noise-shock pairings

and were tested with noise showed lower flight to the noise than mice trained only with shock and then received noise for the first time. Additionally, prior habituation to the noise or experience with the noise during training (i.e. paired and unpaired groups) further reduced noise-elicited flight at test.

Another important factor to consider is the timing of the activity burst with respect to CS and US onset. With poorly timed and sustained conditional responses such as freezing the CR tends to fill the entire CS-US interval and spill over beyond the time of expected US delivery (e.g. *Ayres and Vigorito, 1984*; *Gale et al., 2004*). However, shorter duration ballistic responses such as the darting response allow a clearer assessment of when the CR occurs with respect to CS and US delivery, and such CRs are expected to anticipate US delivery. *Hull, 1934* cautioned conditioning researchers that it is important to distinguish true conditional responses from unconditional responses to the CS, which he named alpha responses. These alpha responses occur at the onset of the CS, rather than the time of the expected US. Alpha responses have been most studied with the Pavlovian conditional eyeblink response, where the true CR is well timed to US delivery (*McCormick and Thompson, 1984*; *Perrett et al., 1993*). Blinks that occur to CS onset are classified as alpha responses, which are considered to be nonassociative startle responses to the CS and not CRs (e.g. *Gerwig et al., 2005*; *Nation et al., 2017*; *Schreurs and Alkon, 1990*, *Woodruff-Pak et al., 1996*). Typically, in eyeblink studies alpha responses are excluded from analysis by omitting any responses that occur at the beginning of the CS. Our darting responses almost exclusively occurred at CS onset and there were rarely any US anticipatory-like responses. Thus, traditional Pavlovian analyses for ballistic CRs would have categorized darting as an unconditional alpha response and not a bona fide CR. Consistent with this analysis is that darting occurred to the noise during the first few trials of the habituation session in Experiment 4 (*Figure 8—figure supplement 1*).

Our interpretation that noise unconditionally elicits a ballistic activity burst bears some relationship to the unconditional acoustic startle response. Loud noises will elicit an unconditional startle response that wanes with repeated presentations of that noise (i.e. habituation; e.g. *Davis, 1980*; *Hoffman and Fleshler, 1963*; *Leaton, 1976*). The unconditional startle response to the loud noise can be potentiated if the loud noise is delivered in the presence of a cue or a context that has been associated with shock/fear (*Brown et al., 1951*; *Davis, 1989*). While our 75 dB noise stimulus is less intense than the 98–120 dB noise used in typical acoustic startle studies, we observed an unconditional noise-elicited response that also decreased with habituation (Experiment 4). Furthermore, our data and those of *Totty et al., 2021* indicate that these responses require a fearful context in order to occur. Fear is well known to potentiate startle responses (*Brown et al., 1951*; *Davis, 1989*). Perhaps the low intensity noise is below threshold to elicit a startle response on its own, but a fearful context potentiates this unconditional startle response and brings it above threshold. Additionally, there appears to be considerable overlap in the neuroanatomy that supports this circa-strike behavior and fear potentiated startle. *Totty et al., 2021* found that inactivation of the central nucleus or the bed nuclei of the stria terminalis disrupts the flight response. These two regions have been shown to be important mediators of fear's ability to potentiate startle (e.g. *Campeau and Davis, 1995*; *Davis and Walker, 2014*). Furthermore, *Fadok et al., 2017* reported that it is corticotropin-releasing hormone (CRH) expressing cells, but not somatostatin expressing cells, within the central nucleus that support flight behavior. Again, there is extensive data implicating CRH and fear potentiated startle (*Lee and Davis, 1997*).

It is of note that the relationship between startle (circa-strike defense) and freezing (post-encounter defense) was described by *Fanselow and Lester, 1988* when accounting for how rats rapidly transitioned between these behaviors when a detected predator launches into attack. "It is as if the freezing animal is tensed up and ready to explode into action if the freezing response fails it. This explosive response probably has been studied in the laboratory for over 30 years under the rubric of potentiated startle… It seems that the releasing stimulus for this explosive motor burst is a sudden change in the stimulus context of an already freezing rat (*Fanselow and Lester, 1988*, p 202)."

Neither *Fadok et al., 2017* nor *Gruene et al., 2015* included any controls for nonassociative behavior, which is something required in order to conclude that a response is conditional (*Rescorla, 1967*). Both of these research groups concluded from their single-group experiments that flight/darting was a CR because the behavior increased with successive shocks during the shock phase and decreased with shock omission during the test phase, likening these behavioral changes to acquisition and extinction. While acquisition and extinction are characteristics of a CR, learning theorists

have never taken these as diagnostic of a CR. For example, increases in responding with successive shocks could arise via sensitization and decreases in responding when shocks are omitted could arise from habituation. Indeed, that is exactly what we believe caused these behavioral changes that we also observed in our study. Shocks, by conditioning fear to the context, sensitize or potentiate the darting response and repeated presentations of the noise alone cause the response to habituate. The behavior of our pseudoconditioning control provides clear evidence of this. Just giving shocks conditioned fear to the context such that when the noise was presented for the first time during test it caused a strong activity burst. The behavior gradually decreased during testing because repeated presentations of the noise led to habituation of this unconditional response.

Given our argument that the flight/darting behavior is nonassociative, Totty et al.'s finding that noise-shock paired rats showed more noise elicited activity burst behavior than rats that had unpaired noise and shock requires additional comment. Since both unpaired and paired rats were exposed to noise during acquisition those exposures could lead to habituation of the unconditional response to the noise. However, it would be expected that habituation would be greater in the unpaired group because pairing a stimulus (noise in this case) with another stimulus (shock in this case) is known to reduce the magnitude of habituation (*Pfautz et al., 1978*). This reduction in habituation is observed even if the second stimulus is not an unconditional stimulus (*Pfautz et al., 1978*). Additionally, pairing a habituated stimulus with a US can also cause a return of the habituated alpha response, and this loss of habituation is not observed when the two stimuli are not paired (*Holland, 1977*). Thus, the difference between the paired and unpaired groups reported by *Totty et al., 2021* is likely due to differential habituation of the noise during training. This effect of habituation was probably enhanced by Totty et al. including a noise habituation phase prior to training.

While many of the findings presented in *Totty et al., 2021* are in line with our own or can be readily explained through a nonassociative lens, Figure 5 of *Totty et al., 2021* presents data in which noise-elicited flight is greater for animals that have SCS-US pairings compared to those that have received only the US during training (comparable to paired vs shock only groups in Experiments 2–4 here). It is possible that Totty et al.'s additional habituation to the SCS in an alternate context prior to SCS or US-only training is a potential explanation for this difference. Indeed, the results here from Experiment 4 in which habituation was conducted in the training/testing context suggest that such habituation would primarily function to reduce darting, particularly in groups that only had US presentations. The use of a serial tone-noise compound may have also served to further reduce a startle response to the noise in Totty et al. It is well known that if a startle stimulus is preceded by another stimulus, the startle response is reduced in a phenomenon called pre-pulse startle inhibition (*Louthan et al., 2020*; *Groves et al., 1974*). Thus, two factors may contribute to Totty et al.'s failure to see darting behavior in the animals that did not receive SCS-shock pairings: the use of habituation prior to conditioning and the use of a two-stimulus compound. Another factor that may be important is species. Overall, the amount of darting seen in the studies using mice (*Fadok et al., 2017*; *Hersman et al., 2020*; the present study) was greater than in the studies using rats (*Gruene et al., 2015*; *Totty et al., 2021*).

Beyond this potential explanation, as mentioned above and detailed below, there are likely at least two different topographical types of locomotion occurring, and such cue-elicited locomotion behaviors in rats in previous studies may preferentially be 'movement' that is part of the freezing suite of behaviors, which include locomotion to an ideal thigmotaxic place to freeze (e.g. *Fanselow and Lester, 1988*). Associative processes would be expected to alter/increase such increased locomotion as it supports/is a part of freezing behavior.

Experimenters that have examined running-like locomotion in fear conditioning situations tend to collapse the behaviors under a single label such as darting, flight, or escape (*Colom-Lapetina et al., 2019*; *Fadok et al., 2017*; *Gruene et al., 2015*; *Mitchell et al., 2021*; *Totty et al., 2021*). These behaviors are then thought of as 'active' behaviors that compete with a 'passive' freezing response (*Fadok et al., 2017*; *Gozzi et al., 2010*; *Gruene et al., 2015*). It is important to recognize that all such movements are not identical and often serve different functions. Here, we will focus on two distinct movements we observed in the present experiments. In several instances, the mice made two movements in response to stimulus presentation. The first was a very high velocity response that occurred to stimulus onset; the other was a slower velocity movement that tended to occur later on during stimulus presentation (*Figure 9C, D and E*). We will discuss these two behaviors within the predatory

imminence framework and suggest that the slower movement is part of the post-encounter freezing module and the faster one is a circa-strike behavior.

## Post-encounter movement

Elsewhere we pointed out that the dominant post-encounter behavior, freezing, is not simply immobility (*Fanselow and Lester, 1988*). Rather, it is an integrated behavior where rats first move to the closest, easily accessible, location appropriate for freezing (*Fanselow and Lester, 1988*). Typically, this location is against a wall, especially a corner with its two walls (*de Oca et al., 2007*; *Grossen and Kelley, 1972*; *Sigmundi, 1997*). In other words, freezing and thigmotaxis constitute an integrated behavioral module (*Fanselow and Lester, 1988*). Indeed, when an especially appropriate freezing location, a dark cave, was available rats moved to the cave and froze more than when it was not (*de Oca et al., 2007*). Therefore, it would be most inappropriate to characterize the movements needed for thigmotaxis as competing with freezing. They are an integral part of freezing. Consistent with this interpretation, the slower second movement after stimulus onset was directed at a corner and once there the animal became immobile. Since these movements are part of the freezing module and shock-associated conditional stimuli are one of the most effective ways to drive the freezing module, it is not surprising that they may be more frequent in animals that had sound-shock pairings (*Figures 4–8*). We conjecture that the movements seen by *Totty et al., 2021* exclusively in animals that were conditioned were largely this type of movement. Further, it is likely that the flight behavior in rats observed by other groups (*Colom-Lapetina et al., 2019*; *Gruene et al., 2015*; *Mitchell et al., 2021*) is this type of movement as Mitchell et al. report that the majority of their CS-elicited darting occurs late in the CS presentation, about 7–10 s into a 30 s tone CS.

## Circa-strike activity bursts

Circa-strike behavior occurs at or immediately before contact with the predator and represents a vigorous evasive movement away from the predator. The most effective circa-strike eliciting stimulus in the laboratory setting is shock, as it is directly analogous to painful contact. As can be seen in *Figure 9A–B*, the movement to shock onset is on average the highest velocity movement we observed. The activity burst to shock is highly protean and poorly directed, sometimes looking as if the animal is bouncing off the walls (*Fanselow, 1982*). If directed at all, it is directed away from the predator and not toward anything in particular. We argue here that when the animal is in the post-encounter fear mode (freezing), the threshold for these bursts of activity shifts such that sudden stimuli that would not normally cause an activity burst now do so. These are the high velocity movements we see to the noise and the tone at stimulus onset (*Figure 9*). These are nonassociative responses as they were greatest in animals that never had the sound paired with shock and decreased with familiarity to the stimulus (*Figures 5, 7 and 8*). *Figure 9A–B* shows the high velocity movement to tone, noise, and shock onset. The difference between the stimuli seems quantitative, such that shock > noise > tone, but note that in the violin plot some noise-elicited bursts are as fast as the fastest shock responses and many of the shock-elicited bursts overlapped in velocity with those triggered by tone. In all cases, these movements seemed protean and undirected.

A final clarifying note is in order regarding stimulus type and how this impacts cue-elicited flight. As described throughout the paper and detailed above, this flight or darting behavior can and does occur to both tone and noise stimuli, regardless of whether they have been paired with shock or not. There is a slight difference in the magnitude of the response to the tone vs the noise, with the noise having a higher proportion of particularly high-velocity movements—on par with reaction to shock at times. We have also noted that the white noise CS is not really a neutral cue and can/does support fear conditioning as if it were a US. While it is possible that these aversive properties of the white noise stimulus enhance cue-elicited flight in frightened animals, it is clearly not required as the tone can elicit similar flight. Rather than thinking about whether a cue needs to procedurally be a CS or a US to support darting, the data support the hypothesis that when an animal is afraid, any sudden change in stimulation can cause them to shift from post-encounter responses, like freezing, toward more circa-strike responses, like undirected flight. Thus, it is a sudden change in stimulation, regardless of the conditional/unconditional properties of the cue, that elicits flight behavior.

An alternative explanation of the altered timing of flight behavior in animals who have had noise-shock pairings is that these animals are more accurately timing the CS-US interval and are showing

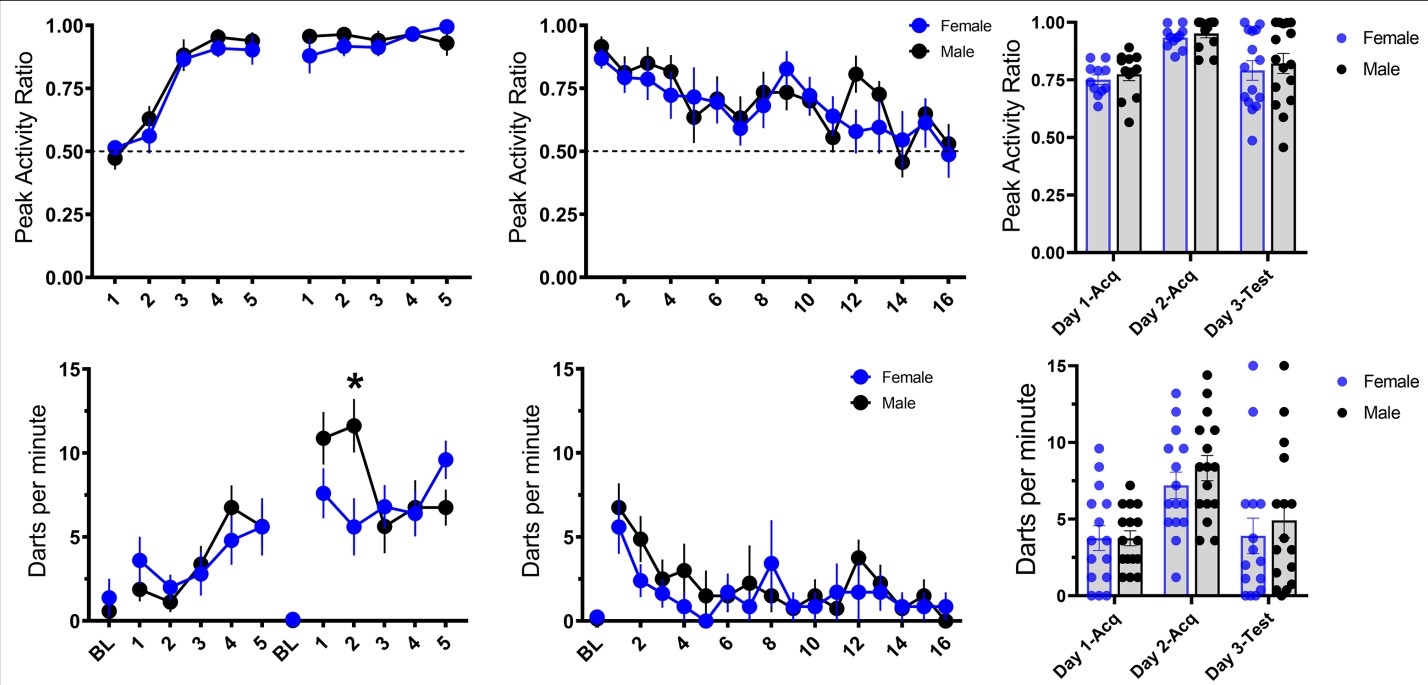

**Figure 11.** Trial-by-trial mean (± SEM), peak activity ratio (PAR), and darting per minute throughout all stimulus presentations during training (left panels), and testing (middle panels) for all groups across experiments that received noise-shock pairings, grouped by sex of the animal (n=15 females; n=16 males). The right panels show individual values for each animal's average PAR and darts per minute across training and testing. p-values and significance were determined through repeated measures ANOVA, and the interaction was followed up with pairwise t-tests. *p<0.05. See *Figure 11—source data 1*.

The online version of this article includes the following source data for figure 11:

**Source data 1.** Source data for the potential sex differences in PAR and darting.

better temporal discrimination and potential avoidance of an upcoming threat. While our data do not necessarily rule out this explanation, the bulk of our results suggests that the majority of the cue-elicited flight behavior observed near cue onset in our experiments was nonassociative in nature. The flight behavior we observed later on in the cue presentation, which may have an associative component, may however be temporally driven. However, it is worth pointing out that this second burst of activity was also not particularly well timed to US delivery, it tended to occur in the middle of the noise presentation (see noise-shock group in *Figure 5*). Future studies that vary the length of the CS in animals who have US-only vs paired CS-US presentations during training are warranted and would help address this alternative explanation.

Initial reports suggest a sex difference in this noise-elicited flight behavior such that female rats show more of this behavior than males (*Gruene et al., 2015*; *Mitchell et al., 2021*). Within each experiment, we found no such sex differences between male and female mice for the PAR and darting measures of flight behavior, and *Totty et al., 2021* similarly found no sex differences in such behavior in male and female rats. To further increase the power of such an analysis for sex differences, we pooled all of the groups across the four experiments that received noise-shock pairings (*Figure 11*). In this analysis, again, we saw no sex differences in flight to the white noise across the 2 days of acquisition for PAR (day 1: $F_{[1, 29]}=0.323$, $p=0.58$; day 2: $F_{[1, 29]}=0.507$, $p=0.48$). For darting, there were no impacts of sex on day 1 ($F_{[1, 29]}=0.009$, $p=0.92$). On day 2, there was a trend for a main effect of sex ($F_{[1, 29]}=3.752$, $p=0.06$) and CS presentation X sex interaction ($F_{[4, 116]}=2.558$, $p=0.042$), such that males darted more than females on the second CS presentation. We further observed no sex differences across testing to the white noise in extinction for both PAR ($F_{[1, 20]}=0.099$, $p=0.76$) and darting ($F_{[1, 13]}=1.397$, $p=0.258$). Thus, we show no major sex differences other than a potential increase in male darting on day 2 of acquisition. Perhaps initial reports of sex differences with more frequent darting in females could be explained by differences in handling and stress provided to females as a result of monitoring estrous phase, a potentially stressful procedure for the animals for which there is

not an ideal control in males. However, results were recently replicated in which females show more darting than males in a sample of animals for which estrous cycle was not monitored (*Mitchell et al., 2021*). It is worth noting that *Gruene et al., 2015*, *Colom-Lapetina et al., 2019*, and *Mitchell et al., 2021*, generally find very low levels of darting, with a majority of animals classified as nondarters. With the procedure used here, and in mice, we generally do not observe different subpopulations of darters vs nondarters (*Figure 11*). While there is certainly variation in the level of darting between animals, all animals across all experiments were shown to dart at least once and would be classified as darters using the criteria set forth in *Gruene et al., 2015*. Finally, while we show no major differences in tone-elicited flight vs noise-elicited flight, we do show some evidence that darts to the tone may be less strong than darts to the noise (*Figure 9A*). Perhaps there are species differences in such stimulus-evoked flight behavior, such that tone-elicited flight in rats is more sensitive to impacts of sex.

Some have characterized freezing as a passive response (*Blanchard and Blanchard, 1969*; *Fadok et al., 2017*; *Gozzi et al., 2010*; *Gruene et al., 2015*; *Yu et al., 2016*) that occurs because no other response is available (*Blanchard, 1997*; *Blanchard et al., 1989*). However, because motion is often the releasing stimulus for predatory attacks it is the best thing for a small mammal like a rat or a mouse to do when a predator is detected and will only be replaced if there is a change consistent with contact (*Fanselow and Lester, 1988*). Rodents choose locations in which to freeze such as corners or objects (thigmotaxis) (*Grossen and Kelley, 1972*). The current data show that the freezing rodent also prepares to react to sudden stimulus change. There is nothing passive about freezing.

## Materials and methods

### Subjects

Subjects for all experiments included 120 C57BL/6NHsd mice (Experiment 1, n=24; Experiment 2, n=32; Experiment 3, n=32; Experiment 4, n=32), aged 9–11 weeks of age and purchased from Envigo. This C57BL/6NHsd strain was chosen to match that of *Fadok et al., 2017*. Each group consisted of four male and four female mice. A necessary/powered group sample size of 8 was calculated based on both years of data in our lab that suggests n=8 is sufficient to detect such behavioral differences in fear conditioning studies and on the recent articles in the literature using this procedure. Mice were group-housed four per cage on a 12 hr light/dark cycle with ad libitum access to food and water. Across each experiment, mice in each cage were randomly assigned to one of the groups, ensuring that every group had a representative from each cage to avoid any cage effects. All experiments were conducted during the lights-on phase of the cycle. Animals were handled for 5 days prior to the start of experiments. Subjects were all treated in accordance with a protocol approved by the Institutional Animal Care and Use Committee at the University of California-Los Angeles following guidelines established by the National Institute of Health.

### Apparatus and stimuli

All experiments were conducted in standard MedAssociates fear conditioning chambers (VFC-008; 30.5 × 24.2 × 21 cm), controlled by Med Associates VideoFreeze software (Med Associates, St. Albans, VT). For each experiment, the same context was used for training and testing (see Discussion). The context was wiped down between each mouse with 70% isopropanol and three sprays of 50% Windex were added to the pans below the shock grid floors to provide an olfactory cue/context. The US consisted of a 1 s 0.9 mA scrambled shock delivered through a MedAssociates shock scrambler (ENV-414S). Each of the CSs was delivered using a MedAssociates speaker (ENV-224AM-2). The tone was 7.5 kHz. Both the tone and the white noise were 75 dB inside the chamber. The intertrial interval varied between 150 s and 210 s with an average length of 180 s.

### Design and procedure

Mice were handled for 5 days for approximately 1 min per day prior to beginning the experiment. At the beginning of each day of the experiment, mice were transported in their home cages on a cart to a room adjacent to the testing room and allowed to acclimate for at least 30 min. Mice were individually placed in clean empty cages on a utility cart for transport from this room to the testing room and promptly returned to their home cages after the session was over. These transport cages were wiped down with StrikeBac in between trials/sessions.

Experiment 1 was conducted as delineated in *Table 1* (see *Figure 1* for a schematic representation of the serial conditioned stimulus and the design for training and testing for Experiment 1). The Replication Group was trained on each of the 2 days with five presentations of a 10 s tone immediately followed by a 10 s noise, which was immediately followed by a 1 s shock. On day 3, it was then tested with 16 presentations of a 10 s tone immediately followed by a 10 s noise. These parameters were chosen to match those of *Fadok et al., 2017* except that we did not include a session of unreinforced CS pre-exposure prior to conditioning as such treatment is known to reduce conditional behavior (*Lubow and Moore, 1959*; we did add such a treatment to Experiment 4 as an experimental factor). The CS Duration Group was trained on each of the 2 days with five presentations of a 10 s noise, which was immediately followed by a 1 s shock. It was tested with 16 presentations of the 10 s noise. The Stimulus Change Group was trained on each of the 2 days with five presentations of a 20 s tone immediately followed by a 1 s shock. It was tested with 16 presentations of a 10 s tone immediately followed by a 10 s noise (i.e. the compound used in the replication group). Two mice were excluded from this study due to experimenter error, one female in the Replication Group and one female in the Stimulus Change Group.

Experiment 2 was conducted as delineated in *Table 2*. The pseudoconditioned noise and pseudoconditioned tone groups were trained on each of the 2 days with five presentations of a 1 s shock without any sound using the same schedule for shocks as Experiment 1. The no shock control was merely allowed to explore the context for the same length of time as the other groups without receiving any shock or auditory stimuli throughout the 2 days of acquisition. The final noise-shock conditioning group was trained on each of the 2 days with five presentations of a 10 s noise, which was immediately followed by a 1 s shock. As Experiment 1 revealed that similar behavior was observed in groups that received compound stimulus-shock pairings or just noise-shock pairings, we used simple noise-shock pairings in this and some of the following experiments to more specifically assess the associative nature of any white-noise-driven behavior. All groups received tests with 16 presentations of the 10 s noise in extinction, except for one of the pseudoconditioning groups that was tested with the 10 s tone.

Experiment 3 was conducted as delineated in *Table 3*. The paired noise-shock (conditioning) group was trained on each of the 2 days with five presentations of a 10 s noise, which was immediately followed by a 1 s shock. The unpaired noise-shock group was presented with the same number and length of noise and shocks, but they were explicitly unpaired in time. The noise-CS only group received five presentations of a 10 s noise without receiving any shocks on each of the 2 days. The shock only (pseudoconditioning) group received five presentations of a 1 s shock on each of the 2 days. As the main behavioral responses and differences between groups occurred primarily in the first few trials of the previous experiments, and in order to more readily complete all of the testing within one day's light cycle, for this and the following experiments we reduced the number of test trials presented to the animals. Thus, at test for this experiment, all groups received two presentations of a 10 s noise.

Experiment 4 was conducted as delineated in *Table 4*. Prior to training with shock, all groups underwent 2 days of additional training with either habituation to the white noise or merely exposure to the context. The habituated groups, habituation/shock only (H-shock) and habituation/noise-shock pairing (H-paired), were trained on each of the 2 days with five presentations of a 10 s noise, while the two nonhabituated groups, context exposure/shock only (C-shock) and context exposure/noise-shock pairing (C-paired) received only equivalent exposure to the context. The following 2 days, as in the experiments above, all groups received 10 footshocks. The paired groups (H-paired and C-paired) were trained on each of the 2 days with five presentations of a 10 s noise, followed immediately by a 1 s footshock. The shock only groups (H-shock and C-shock) were trained on each of the 2 days with only five presentations of a 1 s footshock. At test, all groups received three presentations of the 10 s noise.

## Data, statistics, and analysis

Freezing behavior for Experiments 1–3 was scored using the near-infrared VideoFreeze scoring system. Freezing is a complete lack of movement, except for respiration (*Fanselow, 1980*). VideoFreeze allows for the recording of real-time video at 30 frames per second. With this program, adjacent frames are compared to provide the gray scale change for each pixel, and the amount of pixel change across each frame is measured to produce an activity score. We have set a threshold level of

activity for freezing based on careful matching to hand-scoring from trained observers (**Anagnostaras et al., 2010**). The animal is scored as freezing if they fall below this threshold for at least a 1 s bout of freezing.

For Experiment 4, due to a technical error, videos for the first 4 days of the experiment could not be accurately assessed for freezing behavior using VideoFreeze. Therefore, we alternatively measured and scored freezing behavior using EthoVision. Briefly, videos were converted to MPEG, as described above, and analyzed using the activity analysis feature of Ethovision. Thresholds for freezing were again determined to match hand-scoring from trained observers.

Two different measures of flight were used. We scored bursts of locomotion and jumping with a PAR (**Fanselow et al., 2019**) and the number of darts (**Gruene et al., 2015**). To determine PAR, we took the greatest between frame activity score during a period of interest (e.g. the first 10 s of CS presentation = During) and calculated a ratio of that level of activity to a similar score derived from a preceding control period of equal duration (e.g. 10 s before presentation of the tone = PreStim) of the form During/(During + PreStim). For each CS, the PreStim values were taken from the immediately preceding 10 s period prior to the CS onset. With this measure, a 0.5 indicates that during the time of interest there was no instance of activity greater than that observed during the control period (PreStim). PARs approaching 1.0 indicate an instance of behavior that far exceeded baseline responding. This measure reflects the maximum movement the animal made during the period of interest.

Darting was assessed as in **Gruene et al., 2015**. Video files from VideoFreeze were extracted in Windows Media Video format (.wmv) and then converted to MPEG-2 files using Any Video Converter (AnvSoft, 2018). These converted files were then analyzed to determine animal velocity across the session using EthoVision software (Noldus), using a center-point tracking with a velocity sampling rate of 3.75 Hz. This velocity data was exported, organized, and imported to R (**R. Core Team, 2018**). Using a custom R code (available as **Source code 1**), darts were detected in the trace with a minimum velocity of 22.9 cm/s and a minimum interpeak interval of 0.8 s. The 22.9 cm/s threshold was determined by finding the 99.5th percentile of all baseline velocity data analyzed, prior to any stimuli or shock, and this threshold was validated to match with manual scoring of darts, such that all movements at that rate or higher were consistently scored as darts. See **Figure 1—figure supplement 1** for representative traces of velocity across day 1 of acquisition for a mouse in the Replication Group of Experiment 1. The PAR measure reflects the maximum amplitude of movement, while the dart measure reflects the frequency of individual rapid movements.

Trial-by-trial measures of freezing and flight were analyzed with a repeated measure multifactorial ANOVA and post hoc Tukey tests. Baseline freezing and overall responding were collapsed across session when appropriate and then analyzed with a univariate ANOVA test. To directly compare each groups' activity and the magnitude of any flight behaviors during extinction testing, velocity data was binned into 0.533 s bins and subsequently analyzed using repeated measures ANOVA in R. Whenever violations of sphericity were found, the Greenhouse-Geisser correction was used to produce corrected degrees of freedom and p-values. For analysis of darting magnitude and timing, Welch's ANOVA test was used when assumptions of homogeneity of variance were not met. For comparisons of within-subject dart magnitude, paired sample t-tests were performed. Significant effects and interactions were followed up with simple main effects and Bonferroni-corrected pairwise t-tests. A value of $p<0.05$ was the threshold used to determine statistical reliability. For all experiments described above, no effects of sex were observed in initial comparisons/ANOVAs. Sex was thus removed as a factor in subsequent statistical analyses.

## Acknowledgements

We would like to thank the UCLA behavioral testing core and its supervisors Irina Zhuravka and Lindsay Lueptow for conducting the behavioral experiments found in this article. The work conducted here was supported by an NIH grant R01MH062122 (MSF), by the Staglin Center for Brain and Behavioral Health (MSF), and by a NIDA T32 training grant T32DA024635 (JMT).

## Additional information

### Competing interests
Michael S Fanselow: is a founding board member of Neurovation, Inc. The other authors declare that no competing interests exist.

### Funding

| Funder | Grant reference number | Author |
|---|---|---|
| National Institutes of Health | R01MH062122 | Michael S Fanselow |
| Staglin Center for Brain And Behavioral Health | MSF Award | Michael S Fanselow |
| National Institute on Drug Abuse | T32DA024635 | Jeremy M Trott |

The funders had no role in study design, data collection and interpretation, or the decision to submit the work for publication.

### Author contributions
Jeremy M Trott, Conceptualization, Data curation, Formal analysis, Methodology, Resources, Software, Validation, Visualization, Writing – original draft, Writing – review and editing; Ann N Hoffman, Data curation, Formal analysis, Investigation, Methodology, Visualization, Writing – review and editing; Irina Zhuravka, Investigation, Methodology, Project administration, Writing – review and editing; Michael S Fanselow, Conceptualization, Funding acquisition, Methodology, Project administration, Resources, Supervision, Writing – original draft, Writing – review and editing

### Author ORCIDs
Jeremy M Trott http://orcid.org/0000-0002-7875-3446
Michael S Fanselow http://orcid.org/0000-0002-3850-5966

### Ethics
All animal subjects in each reported study were treated in accordance with an approved protocol (#09-107) from the Institutional Animal Care and Use Committee at the University of California-Los Angeles following recommendations in the Guide for the Care and Use of Laboratory Animals established by the National Institute of Health.

### Decision letter and Author response
Decision letter https://doi.org/10.7554/eLife.75663.sa1
Author response https://doi.org/10.7554/eLife.75663.sa2

## Additional files

### Supplementary files
• Transparent reporting form
• Source code 1. Custom R code for darting analysis.

### Data availability
All data generated or analyzed during this study are included in the manuscript and supporting files. R code to extract darts and produce velocity traces is uploaded as Source Code 1. Source Data Files have been provided for Figures 2, 3, 4, 5, 6, 7, 8, 9, 10, 11 as well as Figure 1-figure supplement 1, Figure 2-figure supplement 1, Figure 4-figure supplement 1, Figure 8-figure supplement 1.

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
