## [Editor Report]

This paper will be of interest to neuroscientists, learning theorists, and clinicians concerned with factors influencing threat-related response selection relevant to fear vs panic. The manuscript describes a group of well-designed experiments that investigate whether flight-like behaviors reported by other groups require associative learning in order to occur. The authors demonstrate that flight-like behaviors observed in these tasks are largely the result of non-associative processes.

---

## [Decision Letter]

**Decision letter after peer review:**

Thank you for submitting your article "Conditional Freezing, Flight and Darting?" for consideration by *eLife*. Your article has been reviewed by 3 peer reviewers, including Mario Penzo as Reviewing Editor and Reviewer #1, and the evaluation has been overseen by Michael Taffe as the Senior Editor. The following individual involved in review of your submission has agreed to reveal their identity: Christopher Cain (Reviewer #2).

Essential revisions:

The individual assessments and recommendations from each of the reviewers are included below. In addition, here we provide you with a brief list of items that we collectively consider to be essential revisions that must be addressed in order for the manuscript to be considered further for publication in *eLife*. We also recommend that you address the individual points raised by each of the reviewers in the public reviews and recommendations for authors, as we consider that addressing them will strengthen the overall quality of the paper. Our requested Essential Revisions are to:

1) Clarify throughout the manuscript that while non-associative processes are a major contributor to flight/darts following fear conditioning, these behavioral responses are also shaped by Pavlovian processes.

2) Discuss contradicting observations from the Totty et al., manuscript where fear-potentiated startle and sensitization where disregarded as major contributors of flight/darts (see comments from Reviewer #2).

3) Include graphs of sex-differences analyses and comment on whether any darter vs non-darter subgroups were observed (see comments from Reviewer #2).

4) Discuss whether the authors' explanation of non-associative "flight" relies on the eliciting stimulus being a US. And how the weak US story fits with critiques of the tone-darting paradigm (since tones are not USs).

5) Discuss and contrast the observations on the different PAR profiles for tone responding seen across Exp1 and Exp2. Side-by-side comparison of the data from these experiments (Stimulus Change Group vs the Replication Group) would be most appropriate.

*Reviewer #1 (Recommendations for the authors):*

1) The behavioral data for the Stimulus Change Group in Figure 2 and the Replication Group in Figure S2 are similar but not identical. It appears that more behavioral discrimination (Tone vs Noise) is seen in the Replication Group compared to the Stimulus Change group. That is, mice show more PAR and darts for the tone in Stimulus Change compared to the Replication group. Statistical comparisons of behavioral differences across the two groups. In my opinion the data suggest that there could be differences in how these responses are executed when completely nonassociative vs when associative processes are at play. In other words, yes, PAR responses can be nonassociative as shown by the authors, but association contributes to more precise behavioral execution. Perhaps, this could be attributed to better temporal discrimination of a looming threat. Such interpretation should also be considered in the text itself.

2) In Line 32 (abstract) the authors mention that running and darting are "entirely" nonassociative. However, because associative processes can shape these responses such claims should be softened.

3) The authors propose that a sudden stimulus change causes circa-strike panic-like behavior in a manner that is proportional to the stimulus intensity and novelty. While the latter aspect has been tested, the stimulus "intensity" aspect remains unexplored. It would be valuable for the authors to directly link stimulus intensity to the magnitude of the panic-like responses.

4) Related to point #1, in some cases, the authors should consider reorganizing the presentation of the data to facilitate direct comparisons when relevant. For example, merging data for Figure 2 and S2 into a single figure alongside a statistical comparison of both experiments will aid visualization and interpretation of the data.

*Reviewer #2 (Recommendations for the authors):*

1. Table 1 indicates 5 test trials when there were 16.

2. Line 120: "..maintained throughout acquisition and extinction testing" – did you mean it is maintained at the beginning of extinction? This is what Fadok reports and your data look similar. But PAR scores decline during extinction (not maintained).

3. The presentation of results (graphs) is confusing at times and doesn't match the flow of the manuscript. For instance, the graphs for the first data discussed are in supplemental, forcing the reader to switch gears to find them. Most of these data are central to the points being raised in the manuscript and should not be in supplemental.

4. The text setting up early experiments and rationale for selecting particular control groups could be clearer. For instance, the CS Duration group addressing "brevity" is really testing whether embedding in a serial compound is necessary for the response to noise-not the duration of the noise or its temporal relation to shock. Line 129: again, this is an unnecessarily confusing setup. "to probe the necessity of the compound we trained a third group…" but the previous CS-duration group already showed that a compound is not necessary. This section ends suggesting the purpose of this group was to probe the necessity of noise-shock pairing. Also, in Experiment 2 the Shock-only/Tone_test group is never discussed.

5. Line 547: For individual trials, is the PAR calculated using the period immediately preceding each stimulus (each trial has its own control period)? Or is a single pre-stimulus period used for all trial calculations (common control period for all trials)?

6. Lines91-92: "metrics used to score flight confounded with pre-noise freezing". This is never explained and could be important. Is this because they use decreases in noise freezing as an index of flight? Please explain in text.

7. In a recent preprint the Shansky lab reproduces greater darting behavior in females that were not swabbed for estrous phase determination. So this likely does not explain why they find sex differences in behavior (lines 430-431).

*Reviewer #3 (Recommendations for the authors):*

This manuscript reports some interesting, highly relevant data and raises many important points. Particularly interesting and useful for the field is the rule about the conditions under which white noise stimuli can transition associative freezing to non-associative flight. The paper could, however, do with a more nuanced treatment of its results. Particularly in the discussion, the authors seem to dismiss outright the possibility that darts/flight-like behavior can occur on an associative basis. This seems like an overly firm conclusion, for the reasons described in the public review. Experiments as well-designed and informative as these deserve a discussion that fully delves into the nuances of the data presented.

---

## [Author Response]

Essential revisions:1) Clarify throughout the manuscript that while non-associative processes are a major contributor to flight/darts following fear conditioning, these behavioral responses are also shaped by Pavlovian processes.

While we continue to state that non-associative processes are the prime driver of much of the noise-elicited flight behavior observed in our study, we have clarified our stance on how associative processes may also shape this behavior, both throughout the text generally and with a dedicated addition to the Discussion section that details how, when using the experimental procedures used here, there may be different types of stimulus-evoked activity and that associative processes alter each of these two functional flight behaviors in distinct ways. To support this, we have added a new Figure 9 analyzing dart timing and topography, with relevant text analyses. Based both on prior literature and on an analysis of darting topography in our experiments here, we suggest that initial ballistic bursts of flight to CS onset are topographically and functionally distinct from subsequent, directed bursts of locomotion which occur later in the CS and may be potentiated by CS-shock pairings. This second burst of locomotion is indeed smaller in our data and we propose that it can be thought of as a behavior that is functionally a part of the freezing suite of behaviors, in which animals locomote to appropriate thigmotaxic places to freeze (Fanselow and Lester, 1988).

2) Discuss contradicting observations from the Totty et al., manuscript where fear-potentiated startle and sensitization where disregarded as major contributors of flight/darts (see comments from Reviewer #2).

We thank the reviewers for the suggestion and for allowing us to further explain differences between our data and previously published work from Totty et al., We have added some text in the discussion to address these differences. In addition to potential species differences, procedural differences such as the use of an SCS and the inclusion of additional habituation may account for such differences. Additionally, in the newly-added section of the discussion on the topography of flight behaviors, we suggest one reason for the potential differences in that the flight patterns observed in Totty et al., with rats, may be more indicative of ‘flight’ behaviors that are better thought of as occurring within the ‘freezing’ module of behaviors, which include any movements to a suitable, thigmotaxic place to freeze.

3) Include graphs of sex-differences analyses and comment on whether any darter vs non-darter subgroups were observed (see comments from Reviewer #2).

We thank the reviewers for this good suggestion and have added such graphs (Figure 11). We have also updated the text regarding these data and included text with a potential analysis of darter vs non-darter subgroups. We indeed do not see evidence of darter vs non-darter subgroups, and we generally such much more darting in mice than previous studies in rats which have found sex differences.

4) Discuss whether the authors' explanation of non-associative "flight" relies on the eliciting stimulus being a US. And how the weak US story fits with critiques of the tone-darting paradigm (since tones are not USs).

We have added some data to one Figure (prior Figure S6, now Figure 10) and a brief section discussing whether stimulus-elicited flight relies on the stimulus properties of the stimulus in question. Certainly, this flight does not rely on the eliciting stimulus being a US, as the tone can and does produce such flight behavior, though perhaps to a lesser extent, than a CS which has properties of a US like the noise; and the greatest stimulus-evoked flight is indeed driven by the shock US itself. However, the idea that certain stimuli are CSs and others are USs is more of a procedural than a real difference. Depending on how the experiment is designed, a CS in one experiment can be used as a US in another experiment and vice versa. Thus, we have tried to clarify our position that it is sudden change in stimulation, regardless of the conditional/unconditional properties of the cue, that elicit flight behavior.

5) Discuss and contrast the observations on the different PAR profiles for tone responding seen across Exp1 and Exp2. Side-by-side comparison of the data from these experiments (Stimulus Change Group vs the Replication Group) would be most appropriate.

We thank the reviewers for this suggestion, and we have done as asked. Figure 2 now also shows the data from the prior Figure S2 (comparing Stimulus Change vs the Replication Group). Direct statistical comparisons of these groups did reveal a slight difference such that PAR and darting to stimuli overall was greater in the Stimulus Change group; however, we did not find any statistically reliable evidence that noise-elicited behavior specifically was altered, making it hard to use this data to make claims about whether noise-shock pairings alter such behavior. Even without a reliable interaction with stimulus type, these data do further suggest that associative processes [from noise-shock pairings] reduce the ability for cues to elicit flight behaviors in fearful animals. However/furthermore, we believe our additional analysis in the discussion regarding flight and the impact of associative processes helps to address these concerns.

Reviewer #1 (Recommendations for the authors):1) The behavioral data for the Stimulus Change Group in Figure 2 and the Replication Group in Figure S2 are similar but not identical. It appears that more behavioral discrimination (Tone vs Noise) is seen in the Replication Group compared to the Stimulus Change group. That is, mice show more PAR and darts for the tone in Stimulus Change compared to the Replication group. Statistical comparisons of behavioral differences across the two groups. In my opinion the data suggest that there could be differences in how these responses are executed when completely nonassociative vs when associative processes are at play. In other words, yes, PAR responses can be nonassociative as shown by the authors, but association contributes to more precise behavioral execution. Perhaps, this could be attributed to better temporal discrimination of a looming threat. Such interpretation should also be considered in the text itself.

We thank the reviewer for pointing out this missed opportunity. We have combined Figure 2 and Figure S2 and presented the relevant statistical comparisons in the text of the results. Indeed, there was some evidence that animals in the Stimulus Change group showed higher PAR/more darting than animals in the Replication Group. This suggests that associative processes may actually be suppressing darting. We have further added significant sections to the results and discussion which addresses the potential for associative processes to change the topography/timing of darting and flight responses. Finally, we have added a brief section commenting on the alternative explanation that the altered timing of flight behavior was due to better temporal discrimination of an upcoming threat.

2) In Line 32 (abstract) the authors mention that running and darting are "entirely" nonassociative. However, because associative processes can shape these responses such claims should be softened.

We agree and have softened this language, here and throughout the text so as not to wholly exclude the role that associative processes have on such flight behavior.

3) The authors propose that a sudden stimulus change causes circa-strike panic-like behavior in a manner that is proportional to the stimulus intensity and novelty. While the latter aspect has been tested, the stimulus "intensity" aspect remains unexplored. It would be valuable for the authors to directly link stimulus intensity to the magnitude of the panic-like responses.

We thank the reviewer for this suggestion and have added some analyses and text in the discussion to address this point. Figure 9 and additions to Figure 10 (old Figure S6) showcase how different intensities of stimuli (tone vs noise vs shock) differentially disrupt freezing, support fear conditioning, and elicit darts in a manner scaled with their stimulus intensity.

4) Related to point #1, in some cases, the authors should consider reorganizing the presentation of the data to facilitate direct comparisons when relevant. For example, merging data for Figure 2 and S2 into a single figure alongside a statistical comparison of both experiments will aid visualization and interpretation of the data.

We thank the reviewer and have done as suggested.

Reviewer #2 (Recommendations for the authors):1. Table 1 indicates 5 test trials when there were 16.

Thank you for pointing out this error, which has now been corrected.

2. Line 120: "..maintained throughout acquisition and extinction testing" – did you mean it is maintained at the beginning of extinction? This is what Fadok reports and your data look similar. But PAR scores decline during extinction (not maintained).

Thank you for the recommendation, we have reworded for clarity.

3. The presentation of results (graphs) is confusing at times and doesn't match the flow of the manuscript. For instance, the graphs for the first data discussed are in supplemental, forcing the reader to switch gears to find them. Most of these data are central to the points being raised in the manuscript and should not be in supplemental.

We thank the reviewer for these good suggestions. We have reorganized some figures, including the prior Figure S2 in the new Figure 2. Pre-CS freezing to the context has been added to any graphs which did not have it. Figure 8—figure supplement 1 (old Figure S5) has been adjusted with a single legend for clarity. The y-axes for all darts per minute measures on graphs have been equalized to assist with comparison.

4. The text setting up early experiments and rationale for selecting particular control groups could be clearer. For instance, the CS Duration group addressing "brevity" is really testing whether embedding in a serial compound is necessary for the response to noise-not the duration of the noise or its temporal relation to shock. Line 129: again, this is an unnecessarily confusing setup. "to probe the necessity of the compound we trained a third group…" but the previous CS-duration group already showed that a compound is not necessary. This section ends suggesting the purpose of this group was to probe the necessity of noise-shock pairing. Also, in Experiment 2 the Shock-only/Tone_test group is never discussed.

We thank the reviewer for these good suggestions. We have adjusted the wording of the Results to make the experimental logic clearer, and we have added text to discuss the Shock Only Tone Test Group.

5. Line 547: For individual trials, is the PAR calculated using the period immediately preceding each stimulus (each trial has its own control period)? Or is a single pre-stimulus period used for all trial calculations (common control period for all trials)?

The PAR is calculated using the period immediately preceding each stimulus, and we have added this to the text for clarity.

6. Lines91-92: "metrics used to score flight confounded with pre-noise freezing". This is never explained and could be important. Is this because they use decreases in noise freezing as an index of flight? Please explain in text.

The reviewer brings up a good point. While there are issues with Totty et al.,’s Flight Ratio, as it is confounded with any pre-CS freezing, we too use a similar measure PAR and are able to detect meaningful changes. Furthermore, we feel that we have sufficiently addressed our different findings with Totty et al., in the discussion, such that this initial phrase is not necessary/does not add to the manuscript. Thus, it has been dropped.

7. In a recent preprint the Shansky lab reproduces greater darting behavior in females that were not swabbed for estrous phase determination. So this likely does not explain why they find sex differences in behavior (lines 430-431).

We have added in some references and information to the text for some of the more recent papers from the Shansky lab. We have additionally added some detail to our discussion of why we were unable to replicate the sex differences, which do not rely on estrous monitoring.

Reviewer #3 (Recommendations for the authors):This manuscript reports some interesting, highly relevant data and raises many important points. Particularly interesting and useful for the field is the rule about the conditions under which white noise stimuli can transition associative freezing to non-associative flight. The paper could, however, do with a more nuanced treatment of its results. Particularly in the discussion, the authors seem to dismiss outright the possibility that darts/flight-like behavior can occur on an associative basis. This seems like an overly firm conclusion, for the reasons described in the public review. Experiments as well-designed and informative as these deserve a discussion that fully delves into the nuances of the data presented.

We thank the reviewer, both for the compliments and for the very valid concerns and suggestions. We have added both new data/analysis and a significant portion to the Discussion section to address the subtleties of our data and propose that much of the data can be explained in a functional framework in which the bulk of the cue-elicited flight observed here is an unconditional response to sudden changes in stimulation when afraid. However, there is a distinct type of movement which occurs later in cue presentation, tends to be enhanced by CS-shock pairings, and may be directed movement towards walls/corners where mice can freeze and be thigmotaxic.